# 2D-OOB: Attributing Data Contribution Through Joint Valuation Framework

**Yifan Sun**[*]
University of Illinois Urbana-Champaign
yifan50@illinois.edu

**Jingyan Shen**[*]
Columbia University
js5544@columbia.edu

**Yongchan Kwon**[†]
Columbia University
yk3012@columbia.edu

## Abstract

Data valuation has emerged as a powerful framework for quantifying each datum's contribution to the training of a machine learning model. However, it is crucial to recognize that the quality of *cells* within a single data point can vary greatly in practice. For example, even in the case of an abnormal data point, not all cells are necessarily noisy. The single scalar score assigned by existing data valuation methods blurs the distinction between noisy and clean cells of a data point, making it challenging to interpret the data values. In this paper, we propose 2D-OOB, an out-of-bag estimation framework for jointly determining helpful (or detrimental) samples as well as the particular cells that drive them. Our comprehensive experiments demonstrate that 2D-OOB achieves state-of-the-art performance across multiple use cases while being exponentially faster. Specifically, 2D-OOB shows promising results in detecting and rectifying fine-grained outliers at the cell level, and localizing backdoor triggers in data poisoning attacks.

## 1 Introduction

From customer behavior prediction and medical image analysis to autonomous driving and policy making, machine learning (ML) systems process ever increasing amounts of data. In such data-rich regimes, a fraction of the samples is often noisy, incorrect annotations are likely to occur, and uniform data quality standards become difficult to enforce. To address these challenges, data valuation emerges as a research field receiving increasing attention, focusing on properly assessing the contribution of each datum to ML training [13, 26]. These methods have proven useful in identifying low-quality samples that can be detrimental to model performance, as well as selecting subsets of data that are representative of enhanced model performance [52, 30, 49]. Furthermore, they are widely applicable in data marketplace for fair revenue allocation and incentive design [55, 48, 43, 5].

Nevertheless, existing data valuation methods assign a scalar score to each datum, thereby failing to account for the varied roles of individual cells. This leaves the valuation rationale unclear and can be unsatisfactory and sub-optimal in various practical scenarios. Firstly, whenever a score is assigned to a data point by a particular data valuation method, it is crucial to understand the underlying justifications to ensure transparency and reliability, especially in high-stakes decision making [42]. Secondly, it is important to recognize the fact that even if a data point is of low quality, it is rarely the case that all the cells within this data point are noisy [40, 29, 46]. The absence of detailed insights

---

[*]Equal contribution.
[†]Corresponding author.

38th Conference on Neural Information Processing Systems (NeurIPS 2024).

| Age | Income | Experience | Education |
|------|--------|------------|-----------|
| 25 | 50000 | 3 | 4 |
| 34 | 62000 | 10 | 6 |
| 45 | -1 | 20 | 8 |
| 29 | 35000 | 5 | 5 |
| 40 | 80000 | 15 | 100 |

(a) Data valuation

| Age | Income | Experience | Education |
|------|--------|------------|-----------|
| 25 | 50000 | 3 | 4 |
| 34 | 62000 | 10 | 6 |
| 45 | -1 | 20 | 8 |
| 29 | 35000 | 5 | 5 |
| 40 | 80000 | 15 | 100 |

low valuation scores

(b) Joint valuation

Figure 1: **Comparison of data valuation and joint valuation.** (a) Data valuation evaluates the quality of individual data points, whereas (b) joint valuation evaluates the quality of individual cells. Both panels illustrate the same hypothetical dataset, while darker colors indicate higher quality or importance. As illustrated in panel (a), data valuation can only identify that the third and fifth data points are of low quality, but it lacks further feature-level attribution. This limitation may result in discarding the entire data point, even when only certain cells are problematic. In contrast, joint valuation provides a finer level of attribution than data valuation and aims to reveal how individual features contribute to data values. As shown in panel (b), the joint valuation framework can identify outlier cells (highlighted by blue boxes), such as $-1$ in "Income" and $100$ in "Education", providing detailed interpretations of data values.

into how individual cells contribute to ML training inevitably leads to discarding the entire data point. This can result in substantial data waste, particularly when only a few cells are noisy and data acquisition is expensive. Finally, in data markets, different cells within a data point may originate from different data sellers [3, 11]. Consequently, a singular valuation for the entire point fails to offer equitable compensation to all contributing parties.

**Our contributions**   In this paper, we propose `2D-OOB`, a powerful and efficient joint valuation framework that can attribute a data point's value to its features. `2D-OOB` quantifies the importance of each *cell* in a dataset, as illustrated in Figure 1, providing interpretable insights into which cells are associated with influential data points. Our method is computationally efficient as well as theoretically supported by its connections with `Data-OOB` [26]. Moreover, our extensive empirical experiments demonstrate the practical effectiveness of `2D-OOB` in various use cases. `2D-OOB` accurately identifies cell outliers and pinpoints which cells to fix to improve model performance. Additionally, `2D-OOB` enables inspection of data poisoning attacks by precisely localizing the backdoor trigger, an artifact inserted into a training sample to induce malicious model behavior [14, 6]. On average, `2D-OOB` is 200 times faster than state-of-the-art methods across all datasets examined.

## 2   Preliminaries

**Notations**   Throughout this paper, we focus on supervised learning settings. For $d \in \mathbb{N}$, we denote an input space and an output space by $\mathcal{X} \subseteq \mathbb{R}^d$ and $\mathcal{Y} \subseteq \mathbb{R}$, respectively. We denote a training dataset with $n$ data points by $\mathcal{D} = \{(x_i, y_i)\}_{i=1}^n$ where $(x_i, y_i)$ is the $i$-th pair of the input covariates $x_i \in \mathcal{X}$ and its output label $y_i \in \mathcal{Y}$. For an event $A$, an indicator function $\mathbb{1}(A)$ is 1 if $A$ is true, otherwise 0. For $j \in \mathbb{N}$, we set $[j] := \{1, \ldots, j\}$. For a set $S$, we denote its power set by $2^S$ and its cardinality by $|S|$.

**DataShapley**   The primary goal of data valuation is to quantify the contribution of individual data points to a model's performance. Leveraging the Shapley value in cooperative game theory [41], `DataShapley` [13] measures the average change in a utility function $U : 2^{\mathcal{D}} \to \mathbb{R}$ over all possible subsets of the dataset that either include or exclude the data point. For $i \in [n]$, `DataShapley` of $i$-th datum is defined as follows.

$$\phi_i^{\mathrm{Shap}} := \frac{1}{n} \sum_{k=1}^{n} \frac{1}{\binom{n-1}{k-1}} \sum_{S \in \mathcal{D}_k^{(i)}} [U(S \cup \{(x_i, y_i)\}) - U(S)], \tag{1}$$

where $\mathcal{D}_k^{(i)} := \{S \subseteq \mathcal{D} | (x_i, y_i) \notin S, |S| = k-1\}$. DataShapley $\phi_i^{\mathrm{Shap}}$ in (1) considers every set $S \in \mathcal{D}_k^{(i)}$ and computes the average difference in utility $U(S \cup \{(x_i, y_i)\}) - U(S)$. It characterizes the impact of a data point, but its computation requires evaluating $U$ for all possible subsets of $\mathcal{D}$, rendering precise calculations infeasible. Many efficient computation algorithms have been proposed [16, 28, 54], and in these studies, Shapley-based methods have demonstrated better effectiveness in detecting low-quality samples than standard attribution approaches, such as leave-one-out and influence function methods [22, 10].

**Data-OOB** As an alternative efficient data valuation method, Kwon and Zou [26] propose `Data-OOB`, which leverages a bagging model and measures the similarity between the nominal label and weak learners' predictions. To be more specific, consider a bagging model consisting of $B$ weak learners, where for $b \in [B]$, the $b$-th weak learner $\hat{h}_b$ is given as a minimizer of the weighted empirical risk,

$$\hat{h}_b := \mathrm{argmin}_h \sum_{i=1}^n w_{bi} \ell(y_i, h(x_i)),$$

where $\ell : \mathcal{Y} \times \mathcal{Y} \to \mathbb{R}$ is a loss function and $w_{bi} \in \mathbb{N}$ is the number of times the $i$-th datum $(x_i, y_i)$ is selected by the $b$-th bootstrap dataset. Let $\mathbf{w}_b$ be a weight vector $\mathbf{w}_b := (w_{b1}, \ldots, w_{bn})$ for all $b \in [B]$. For $i \in [n]$ and $\{(\mathbf{w}_b, \hat{h}_b)\}_{b=1}^B$, `Data-OOB` of the $i$-th datum is defined as follows.

$$\phi_i^{\mathrm{OOB}} := \frac{\sum_{b=1}^B \mathbb{1}(w_{bi} = 0) T(y_i, \hat{h}_b(x_i))}{\sum_{b=1}^B \mathbb{1}(w_{bi} = 0)}, \tag{2}$$

where $T(y_i, \hat{h}_b(x_i))$ is a score function evaluated at $(x_i, y_i)$. We assume that the higher $T$, the better the prediction. In classification settings, a common choice for $T$ is $\mathbb{1}(y_i = \hat{h}_b(x_i))$, and in this case, `Data-OOB` $\phi_i^{\mathrm{OOB}}$ measures the average similarity between a nominal label $y_i$ and weak learners' predictions $\hat{h}_b(x_i)$ when a datum $(x_i, y_i)$ is *not* sampled in a bootstrap dataset. It intuitively captures the quality of a data point. For instance, when $(x_i, y_i)$ is a mislabeled sample or an outlier, the label $y_i$ is likely to differ from $\hat{h}_b(x_i)$, resulting in $\phi_i^{\mathrm{OOB}}$ being close to zero.

It is noteworthy that `Data-OOB` in (2) can be computed by training a single bagging model, making it computationally efficient. Kwon and Zou [26] show that `Data-OOB` can easily scale to millions of data points, but for `DataShapley` this is often very impractical. In addition, `Data-OOB` is typically comparable to or even more effective than `DataShapley` in detecting mislabeled data points and selecting helpful data points [26, 17].

## 3 Attributing Data Contribution through Joint Valuation Framework

Data valuation quantifies the utility of data points, however, it fails to identify which features contribute to these data values and to what extent. For instance, in anomaly detection tasks, data valuation methods can be deployed to detect anomalous data points but cannot explain why they are considered abnormal, which is generally not desirable in practice. To address this challenge, we introduce a joint valuation framework that assigns *a cell score* to each feature of a data point. Here, a cell score is designed to quantify how each feature affects the value of an individual data point, thereby attributing the data value to specific features.

To the best of the author's knowledge, Liu et al. [32] were the first to consider the concept of joint valuation in the literature, proposing `2D-Shapley` as a means to quantitatively interpret `DataShapley`. To formalize this, we denote a 2D utility function by $u : [n] \times [d] \to \mathbb{R}$, which takes as input a subset of data points $S \subseteq [n]$ and a subset of features $F \subseteq [d]$, measuring the utility of a fragment of the given dataset consisting of cells $\{(i, j)\}_{i \in S, j \in F}$, where a tuple $(i, j)$ denotes a cell at the $i$-th datum and the $j$-th column. Then, `2D-Shapley` is defined as

$$\psi_{ij}^{\mathrm{2D-Shap}} := \frac{1}{nd} \sum_{k=1}^n \sum_{l=1}^d \frac{1}{\binom{n-1}{k-1}\binom{d-1}{l-1}} \sum_{(S,F) \in \mathcal{D}_{k,l}^{(i,j)}} M_u^{i,j}(S, F) \tag{3}$$

where $\mathcal{D}_{k,l}^{(i,j)} := \{(S, F) | S \subseteq [n] \backslash \{i\}, F \subseteq [d] \backslash \{j\}, |S| = k-1, |F| = l-1\}$ and

$$M_u^{i,j}(S, F) = u(S \cup \{i\}, F \cup \{j\}) + u(S, F) - u(S \cup \{i\}, F) - u(S, F \cup \{j\}).$$

The function $M_u^{i,j}$ allows us to quantify how much removing a specific cell at $(i,j)$ from a given set $(S \cup \{i\}, F \cup \{j\})$ affects the overall utility, and 2D-Shapley $\psi_{ij}^{2D-Shap}$ evaluates the average $M_u^{i,j}$ across all possible data fragments $(S, F) \in \mathcal{D}_{k,l}^{(i,j)}$.

Similar to DataShapley, the permutation of all rows and columns required for exact 2D-Shapley calculations presents significant computational challenges. To address this, Liu et al. [32] develop 2D-KNN, which utilizes $k$-nearest-neighbors models as surrogates to approximate 2D-Shapley values. However, the approximation methods can compromise the accuracy of valuations [26, 17]. Additionally, 2D-KNN still faces challenges scaling to large-scale datasets and high-dimensional settings.

To address these limitations, we propose 2D-OOB, an *efficient* and *model-agnostic* joint valuation framework that leverages out-of-bag estimation to attribute data contribution. We also illustrate how 2D-OOB is connected to Data-OOB, thereby facilitating sample-wise feature-level interpretation for data valuation, as discussed in Section 3.2.

### 3.1 2D-OOB: an efficient joint valuation framework

Our idea builds upon the subset bagging model [15], which is well recognized as an earlier version of Breiman's random forest model [4]. A key distinction from a standard bagging model is that a weak learner in a subset bagging model is trained on a randomly selected subset of features. For $b \in [B]$, we denote the $b$-th random feature subset by $S_b \subseteq [d]$. Then, the $b$-th weak learner of a subset bagging model is given as follows.

$$\hat{f}_b := \operatorname{argmin}_f \sum_{i=1}^n w_{bi} \ell(y_i, f(x_{i,S_b})),$$

where $x_{i,S_b}$ is a subvector of $x_i$ that only takes elements in a subset $S_b$. This difference enables us to assess the impact of which features are more influential: if $S_b$ includes a helpful (or detrimental) feature, we can expect the out-of-bag prediction $\hat{f}(x_{i,S_b})$ to be good (or poor). We formalize this intuition and propose 2D-OOB. For $i \in [n]$, $j \in [d]$ and $\{(\mathbf{w}_b, S_b, \hat{f}_b)\}_{b=1}^B$, the 2D-OOB for the $j$-th cell of the $i$-th data point is defined as follows,

$$\psi_{ij}^{2D-OOB} := \frac{\sum_{b=1}^B \mathbb{1}(w_{bi} = 0, j \in S_b) T(y_i, \hat{f}_b(x_{i,S_b}))}{\sum_{b=1}^B \mathbb{1}(w_{bi} = 0, j \in S_b)}, \tag{4}$$

where $T : \mathcal{Y} \times \mathcal{Y} \to \mathbb{R}$ is a utility function that scores the performance of the weak learner $\hat{f}_b(x_{i,S_b})$ on the $i$-th datum $(x_i, y_i)$. Specifically, for binary or multi-class classification problems, we can adopt $T(y_i, \hat{f}_b(x_{i,S_b})) = \mathbb{1}(y_i = \hat{f}_b(x_{i,S_b}))$. In this case, 2D-OOB measures the average accuracy score of out-of-bag predictions (specifically, when the $i$-th data point is out-of-bag) if the $j$-th feature is used in training $\hat{f}_b$. For regression problems, we can use the negative squared error loss function, defined as $T(y_i, \hat{f}_b(x_{i,S_b})) = -(y_i - \hat{f}_b(x_{i,S_b}))^2$. In practice, $\mathcal{X}$ could also be incorporated into $T$ to suit the specific use case.

While Data-OOB in (2) aims to assess the impact of the $i$-th datum, 2D-OOB in (4) further provides interpretable insights by evaluating the data point with various combinations of features, revealing which cells are influential to model performance. By leveraging the subset bagging scheme, 2D-OOB only requires a single training of the bagging model, making it computationally efficient.

### 3.2 Connection to Data-OOB

We now present interpretable expressions of how 2D-OOB connects to Data-OOB in the following proposition. To begin with, we denote a set of subsets of $[d]$ by $\mathcal{S} := \{S \subseteq [d]\}$. With $\{(\mathbf{w}_b, \hat{f}_b)\}_{b=1}^B$, we define the $i$-th Data-OOB when a particular subset $S$ is used as follows and denote it by $\phi_i^{OOB}(S)$.

$$\phi_i^{OOB}(S) := \frac{\sum_{b=1}^B \mathbb{1}(w_{bi} = 0) T(y_i, \hat{f}_b(x_{i,S}))}{\sum_{b=1}^B \mathbb{1}(w_{bi} = 0)}.$$

**Proposition 3.1.** *For all $i \in [n]$ and $j \in [d]$, $\psi_{ij}^{2D-OOB}$ can be expressed as follows.*

$$\psi_{ij}^{2D-OOB} = \mathbb{E}_{\hat{F}_S}[\phi_i^{OOB}(S) \mid j \in S],$$

*where $\hat{F}_S$ is an empirical distribution with respect to $S$ induced by the sampling process.*

A proof is given in Appendix C. Proposition 3.1 shows that `2D-OOB` $\psi_{ij}^{\text{2D}-\text{OOB}}$ can be expressed as a conditional empirical expectation of `Data-OOB` provided that the $j$-th feature is used in `Data-OOB` computation. It provides intuitive interpretations: for a fixed $i$ and $j \neq k$, $\psi_{ij}^{\text{2D}-\text{OOB}} > \psi_{ik}^{\text{2D}-\text{OOB}}$ implies that the cell $x_{ij}$ is more helpful to achieve a higher OOB score than the cell $x_{ik}$, where the OOB score serves as an indicator of model performance. By distinguishing the contributions of individual cells, `2D-OOB` effectively realizes joint valuation, providing a finer granularity of analysis that links feature-level importance to individual data quality.

## 4 Experiments

In this section, we empirically show the effectiveness of `2D-OOB` across multiple use cases of the joint valuation: *cell-level outlier detection*, *cell fixation*, and *backdoor trigger detection*. To the best of our knowledge, the latter two use cases are introduced here for the first time within the joint valuation framework. As a summary, `2D-OOB` can precisely identify anomalous cells that should be prioritized for examination and subsequent fixation to improve model performance. In the context of backdoor trigger detection, `2D-OOB` demonstrates its efficacy by accurately identifying different types of triggers within poisoned data, showcasing its proficiency in detecting non-random, targeted anomalies. Our method also exhibits high computational efficiency through run-time comparison.

Throughout all of our experiments, `2D-OOB` uses a subset bagging model with $B = 1000$ decision trees. We randomly select a fixed ratio of features to build each decision tree. Unless otherwise specified, we utilize half of the features for each weak learner and set $T(y_i, \hat{f}(x_{i,S_b})) = \mathbb{1}(y_i = \hat{f}(x_{i,S_b}))$. The run time is measured on a single Intel Xeon Gold 6226 2.9 GHz CPU processor. We provide a detailed ablation study of key hyperparameters in Section 4.4.

### 4.1 Cell-level outlier detection

**Experimental setting**   In practical situations, even when dealing with abnormal data points, it is not always the case that all cells are noisy [40, 32, 23]. To simulate more realistic settings, we introduce noise to certain *cells* in the following two-step process: First, we randomly select $20\%$ rows for each dataset. We then select $20\%$ columns uniformly at random, allowing each selected row to have a different set of perturbed cells. We inject noises sampled from the low-probability region into these cells, following Du et al. [9] and Liu et al. [32]. Details on the outlier injection process can be found in Appendix A.3.

We use 12 publicly accessible binary classification datasets from OpenML, encompassing a range of both low and high-dimensional datasets, which have been widely used in the literature [13, 25, 26]. Details on these datasets are presented in Appendix A.1. For each dataset, 1000 and 3000 data points are randomly sampled for training and test datasets, respectively. For the baseline method, we consider `2D-KNN`, a fast and performant variant of `2D-Shapley` [32]. We incorporate a distance regularization term in the utility function $T$ for enhanced performance.

**Results**   We calculate the valuations for each cell using our joint valuation framework. Ideally, the outlier cells should receive a low valuation. We then arrange the cell valuations in *ascending* order and inspect those cells with the lowest values first.

The detection rate curve of inserted outlier is shown in Figure 2. For all datasets, `2D-OOB` successfully identifies over $90\%$ of the outlier cells by inspecting only $30\%$ of the bottom cells. In comparison, `2D-KNN` requires examining nearly $90\%$ of the cells to achieve the same detection level.

We also evaluate the area under the curve (AUC) as a quantitative metric and measure the run-time. As Table 1 shows, `2D-OOB` achieves an average AUC of $0.83$ across 12 datasets, compared to $0.67$ for `2D-KNN`, while being significantly faster. For high-dimensional datasets such as the musk dataset, which comprises 166 features, `2D-KNN` would take more than an hour to process, while `2D-OOB` can finish in seconds. Furthermore, we present additional results on **multi-class classification** datasets in Appendix B.1, demonstrating the consistently superior performance and efficiency of `2D-OOB`.

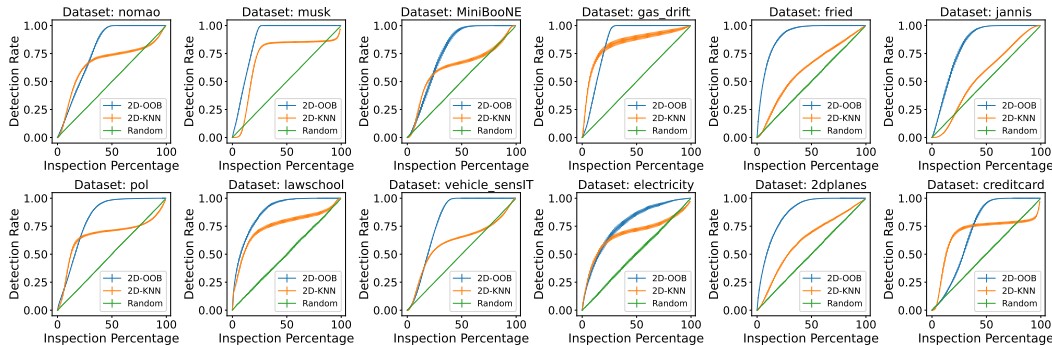

Figure 2: **Cell-level outlier detection rate curves for** `2D-OOB`, `2D-KNN`, **and** `Random`. The x-axis represents the percentage of inspected cells. The y-axis represents the detection rate, defined as the ratio of the number of detected outlier cells to the total number of outlier cells present in a dataset. The error bars show a 95% confidence interval based on 30 independent experiments. We examine the cells in ascending order, starting from those with the lowest values, and thus a curve closer to the left-top corner indicates better performance. `2D-OOB` efficiently detects the majority of outlier cells by examining only a small fraction of the total cells, while `2D-KNN` and `Random` require scanning nearly all the cells.

Table 1: **Cell-level outlier detection results.** AUC and run-time comparison between `2D-OOB` and `2D-KNN` across twelve binary classification datasets. The average and standard error of the AUC and run-time (in seconds) based on 30 independent experiments are denoted by "average ± standard error". Bold numbers denote the best method. The AUC value for the `Random` method consistently remains at 0.5 across all datasets. In every dataset, `2D-OOB` achieves a significantly higher AUC while being orders of magnitude faster than `2D-KNN`.

| Dataset | AUC ↑ | | Run-time ↓ | |
|---|---|---|---|---|
| | 2D-OOB (ours) | 2D-KNN | 2D-OOB (ours) | 2D-KNN |
| lawschool | **0.88± 0.0027** | 0.75± 0.0011 | **3.33 ± 0.06** | 177.56 ± 1.92 |
| electricity | **0.77± 0.0072** | 0.68± 0.0014 | **3.39 ± 0.07** | 191.38 ± 2.60 |
| fried | **0.91± 0.0015** | 0.61± 0.0005 | **3.97 ± 0.10** | 322.79 ± 2.98 |
| 2dplanes | **0.87± 0.0015** | 0.62± 0.0005 | **3.46 ± 0.05** | 295.25 ± 2.37 |
| creditcard | **0.72± 0.0028** | 0.69± 0.0011 | **4.56 ± 0.10** | 662.34 ± 7.12 |
| pol | **0.82± 0.0014** | 0.67± 0.0006 | **4.34 ± 0.05** | 759.33 ± 4.37 |
| MiniBooNE | **0.77± 0.0058** | 0.63± 0.0019 | **7.46 ± 0.06** | 1507.83 ± 14.50 |
| jannis | **0.83± 0.0042** | 0.55± 0.0004 | **7.98 ± 0.07** | 1753.10 ± 12.35 |
| nomao | **0.79± 0.0021** | 0.67± 0.0009 | **7.69± 0.11** | 2564.58 ± 23.11 |
| vehicle_sensIT | **0.81± 0.0014** | 0.61± 0.0005 | **9.87 ± 0.08** | 3113.65 ± 24.54 |
| gas_drift | **0.86± 0.0010** | 0.84± 0.0017 | **11.28± 0.10** | 3878.31 ± 40.72 |
| musk | **0.88± 0.0008** | 0.71± 0.0006 | **14.09 ± 0.11** | 4415.45 ± 22.96 |
| Average | **0.83** | 0.67 | **6.78** | 1636.80 |

## 4.2 Cell fixation experiment

**Experimental setting** A naive strategy to handle cell-level outliers is to eliminate data points that contain outliers. This method, however, risks substantial data loss, particularly when outliers are scattered and data points are costly to collect. We instead consider a cell fixation experiment, where we assume that the ground-truth annotations of outlier cells can be restored with external expert knowledge. At each step, we "fix" a certain number of cells by substituting them with their ground-truth annotations, prioritizing cells that have the lowest valuations. Then we fit a logistic model[3] and evaluate the model's performance with a test set of 3000 samples. It is important to note that correcting normal cells has no effect, whereas fixing outlier cells is expected to enhance the model's performance. We adopt the same datasets and implementations as in Section 4.1.

---

[3]Logistic regression is chosen because it is a simple yet powerful machine learning model commonly used to test data separability, which is a standard practice in this field [17, 32].

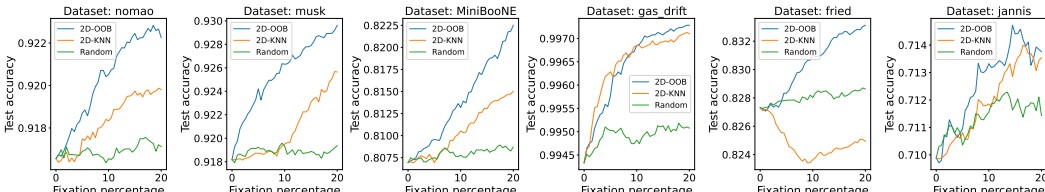

Figure 3: **Cell fixation experiment results (test accuracy curves) for** `2D-OOB`**,** `2D-KNN`**, and** `Random`**.** We replace cells with their ground-truth annotations, starting with those cells assigned the lowest valuations. The results for 6 datasets are presented, and additional results for other datasets are available in Appendix B.2. We conduct 30 independent trials and report the average results. A higher curve indicates better performance. `2D-OOB` demonstrates a superior capability in accurately identifying and rectifying cell-level outliers.

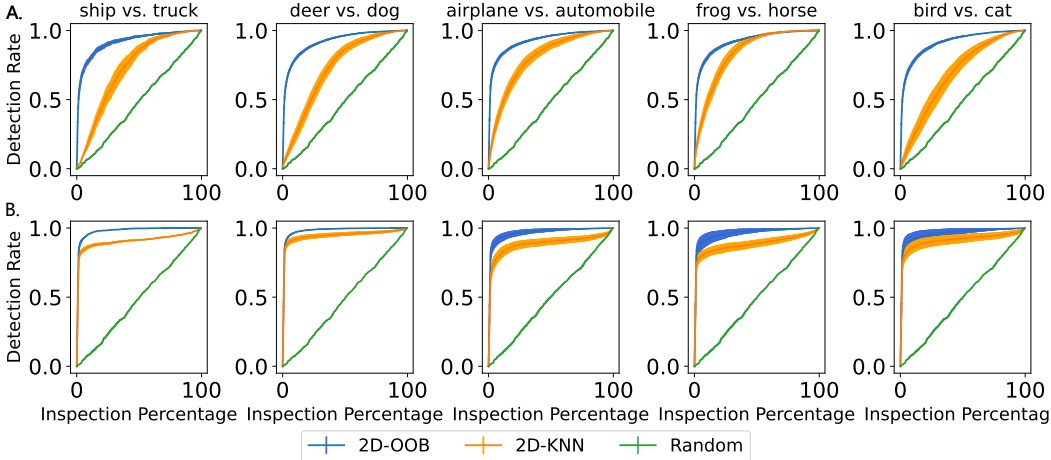

Figure 4: **Backdoor trigger detection rate curves for** `2D-OOB`**,** `2D-KNN`**, and** `Random`**.** Panels A (top) and B (bottom) correspond to the Trojan square and BadNets square, respectively. We inspect the cells within each poisoned sample in descending order of their valuation scores. The detection rate curve shows the average detection rate across all poisoned samples, with error bars representing a $95\%$ confidence interval based on $15$ independent runs. `2D-OOB` demonstrates superior performance in detecting the cells implanted with triggers.

**Results** Figure 3 illustrates the anticipated trend in the performance of `2D-OOB`, validating our method's capability to accurately identify and prioritize the most impactful outliers for correction. As cells with the lowest valuations are progressively fixed, `2D-OOB` demonstrates a consistent improvement in model accuracy. In contrast, when applying the same procedure with `2D-KNN`, such notable performance enhancements are not observed.

Additionally, we investigate a scenario where ground-truth annotations remain unavailable. We adopt the setup from Liu et al. [32], where we replace the outlier cells with the average of other cells in the same feature column. `2D-OOB` uniformly demonstrates significant superiority over its counterparts. Results are provided in Appendix B.2.

### 4.3 Backdoor trigger detection

A common strategy of data poisoning attacks involves inserting a predefined trigger (e.g., a specific pixel pattern in an image) into a subset of the training data [14, 6, 31]. These malicious manipulations can be challenging to detect as they only infect specific, targeted samples. Even when poisoned data are present, it could be difficult to discern the root cause of the attacks, since manually reviewing the images is expensive and time-consuming. In this experiment, we introduce a novel task enabled by the joint valuation framework: localizing backdoor triggers in data poisoning attacks. Distinct from the random outliers investigated previously, this type of cell contamination is targeted and deliberate.

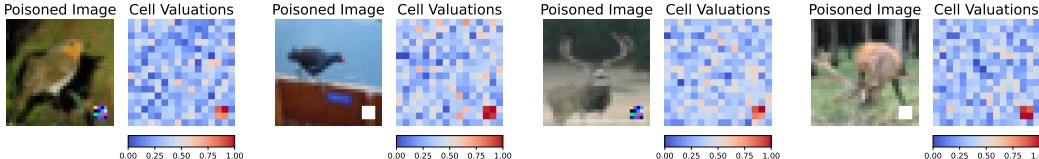

Figure 5: **Qualitative examples for** `2D-OOB` **in the backdoor trigger detection task.** Each pair of images consists of a poisoned image and its corresponding cell valuation heatmap. The color of the heatmap indicates importance, with red cells representing higher importance and blue cells representing lower importance. In the first two pairs, the class 'bird" is relabeled as "cat", while in the latter two pairs, the class "deer" is relabeled as "cat". The heatmaps clearly show that higher cell valuations are predominantly concentrated in the regions containing triggers, while areas featuring actual objects receive lower valuations. This pattern suggests that `2D-OOB` effectively captures the triggers as the impactful features responsible for the misclassification of the poisoned samples.

We consider two popular backdoor attack algorithms: BadNets [14] and Trojan Attack [31]. During training, the poisoned samples, relabeled as the adversarial target class, are mixed up with clean data. As a result, the model learns to incorrectly associate the trigger with the target class. At test time, inputs containing the trigger are misclassified to the target class. In this context, our goal is to effectively pinpoint the triggers by recognizing them as influential features through our joint valuation framework.

**Experimental setting** We select 5 pairs of classes from CIFAR-10 [24]. For each pair, one class is designated as the target attack class, while the other serves as the source class. The training dataset comprises 1000 images. For each attack, we contaminate 15% of the training samples from the source class and relabel them to the target class. Two types of attack triggers are implemented: the Trojan square and the BadNets square [14, 38, 31]. These triggers are placed in the lower right corner of the original images to minimize occlusion. Further details about these attacks are available in Appendix A.4. In our experiment, the ratio of poisoned cells is approximately 1%, and each weak learner in the subset bagging model is built by sampling 25% of the features.

**Results** We adopt the same detection scheme and baseline methods as in Section 4.1. Ideally, the poisoned cells should receive high valuation scores given that such data points have been relabeled. We present the detection rate curves for five datasets in Figure 4. `2D-OOB` significantly outperforms `2D-KNN` in detecting both types of triggers. Overall, `2D-OOB` achieves an average detection AUC of 0.95 across all datasets and attack types, compared to 0.83 for `2D-KNN`. It is worth noting that conventional data valuation methods are fundamentally unable to localize backdoor triggers; at most, they can only identify poisoned data points.

**Qualitative examples** Figure 5 displays the heatmaps for poisoned samples based on cell valuations of `2D-OOB`. Areas with higher cell valuations (marked in dark red) precisely indicate the trigger locations within these samples, demonstrating the effectiveness of our joint valuation framework. Additional examples can be found in Appendix B.3.

## 4.4 Ablation study

We conduct ablation studies on the cell-level outlier detection task, as outlined in Section 4.1, to examine the impact of key hyperparameters on `2D-OOB` estimations, including the selection and number of weak learners, as well as the feature subset ratio.

**Selection of weak learners** Although our study primarily employs decision trees as weak learners, it is important to note that `2D-OOB` is **model-agnostic**, enabling the use of any class of machine learning models as weak learners. Specifically, we examine decision trees, logistic regression, a single-layer MLP with 64 units, and a two-layer MLP with 64 and 32 units, respectively, as weak learners to compare their performance.

Table 2 presents a comparison of detection AUC across 12 datasets with different choices of weak learners, indicating that `2D-OOB` is not model-free. The selection of weak learners *slightly* affects the

Table 2: **Ablation study results of weak learner types.** The average and standard error of the detection AUC based on 30 independent experiments are denoted by "average $\pm$ standard error". Results from 2D-KNN are included for comparison. The choice of weak learner leads to variations in cell values, yet the performance of the detection task remains robust.

| Dataset | Decision Tree | Logistic Regression | MLP (single-layer) | MLP (two-layer) | 2D-KNN (Baseline) |
|---|---|---|---|---|---|
| lawschool | **0.88 $\pm$ 0.0027** | 0.81 $\pm$ 0.0014 | 0.83 $\pm$ 0.0023 | 0.86 $\pm$ 0.0049 | 0.75 $\pm$ 0.0011 |
| electricity | **0.77 $\pm$ 0.0072** | 0.75 $\pm$ 0.0029 | 0.75 $\pm$ 0.0039 | 0.74 $\pm$ 0.0064 | 0.68 $\pm$ 0.0014 |
| fried | **0.91 $\pm$ 0.0015** | 0.82 $\pm$ 0.0023 | 0.85 $\pm$ 0.0020 | 0.88 $\pm$ 0.0027 | 0.61 $\pm$ 0.0005 |
| 2dplanes | 0.87 $\pm$ 0.0015 | 0.82 $\pm$ 0.0026 | 0.86 $\pm$ 0.0026 | **0.88 $\pm$ 0.0037** | 0.62 $\pm$ 0.0005 |
| creditcard | 0.72 $\pm$ 0.0028 | **0.74 $\pm$ 0.0023** | **0.74 $\pm$ 0.0026** | **0.74 $\pm$ 0.0071** | 0.69 $\pm$ 0.0011 |
| pol | 0.82 $\pm$ 0.0014 | 0.79 $\pm$ 0.0029 | 0.85 $\pm$ 0.0014 | **0.86 $\pm$ 0.0019** | 0.67 $\pm$ 0.0006 |
| MiniBooNE | 0.77 $\pm$ 0.0058 | 0.77 $\pm$ 0.0059 | 0.80 $\pm$ 0.0057 | **0.81 $\pm$ 0.0119** | 0.63 $\pm$ 0.0019 |
| jannis | **0.83 $\pm$ 0.0042** | 0.76 $\pm$ 0.0040 | 0.79 $\pm$ 0.0048 | 0.80 $\pm$ 0.0108 | 0.55 $\pm$ 0.0004 |
| nomao | 0.79 $\pm$ 0.0021 | 0.82 $\pm$ 0.0012 | **0.83 $\pm$ 0.0010** | **0.83 $\pm$ 0.0017** | 0.67 $\pm$ 0.0009 |
| vehicle-sensIT | 0.81 $\pm$ 0.0014 | 0.81 $\pm$ 0.0026 | 0.80 $\pm$ 0.0025 | **0.82 $\pm$ 0.0037** | 0.61 $\pm$ 0.0005 |
| gas-drift | 0.86 $\pm$ 0.0010 | **0.89 $\pm$ 0.0005** | 0.88 $\pm$ 0.0005 | 0.88 $\pm$ 0.0006 | 0.84 $\pm$ 0.0017 |
| musk | 0.88 $\pm$ 0.0008 | 0.87 $\pm$ 0.0005 | **0.88 $\pm$ 0.0005** | **0.88 $\pm$ 0.0008** | 0.71 $\pm$ 0.0006 |
| Average | **0.83** | 0.80 | 0.82 | **0.83** | 0.67 |

Table 3: **Ablation study results of (a) the number of base learners $B$ and (b) the feature subset ratio $K/d$.** The average and standard error of the detection AUC based on 30 independent runs are denoted by "average $\pm$ standard error." (a) Increasing the number of base learners from 1000 to 3000 does not yield a notable performance improvement. (b) Our method's joint valuation capacity remains relatively stable regardless of the selected feature subset ratio.

(a) **Ablation on the number of base learners $B$.**

| Dataset | AUC $\uparrow$ | | |
|---|---|---|---|
| | $B = 500$ | $B = 1000$ | $B = 3000$ |
| lawschool | 0.86 $\pm$ 0.0035 | 0.88 $\pm$ 0.0027 | 0.88 $\pm$ 0.0026 |
| electricity | 0.77 $\pm$ 0.0062 | 0.77 $\pm$ 0.0072 | 0.77 $\pm$ 0.0070 |
| fried | 0.87 $\pm$ 0.0022 | 0.91 $\pm$ 0.0015 | 0.91 $\pm$ 0.0014 |
| 2dplanes | 0.87 $\pm$ 0.0016 | 0.87 $\pm$ 0.0015 | 0.87 $\pm$ 0.0015 |
| creditcard | 0.72 $\pm$ 0.0025 | 0.72 $\pm$ 0.0028 | 0.72 $\pm$ 0.0028 |
| pol | 0.78 $\pm$ 0.0022 | 0.82 $\pm$ 0.0014 | 0.82 $\pm$ 0.0014 |
| MiniBooNE | 0.77 $\pm$ 0.0042 | 0.77 $\pm$ 0.0058 | 0.77 $\pm$ 0.0058 |
| jannis | 0.78 $\pm$ 0.0045 | 0.83 $\pm$ 0.0042 | 0.83 $\pm$ 0.0039 |
| nomao | 0.79 $\pm$ 0.0018 | 0.79 $\pm$ 0.0021 | 0.79 $\pm$ 0.0020 |
| vehicle_sensIT | 0.80 $\pm$ 0.0021 | 0.81 $\pm$ 0.0014 | 0.81 $\pm$ 0.0014 |
| gas_drift | 0.86 $\pm$ 0.0007 | 0.86 $\pm$ 0.0010 | 0.86 $\pm$ 0.0010 |
| musk | 0.88 $\pm$ 0.0008 | 0.88 $\pm$ 0.0008 | 0.88 $\pm$ 0.0008 |
| Average | 0.81 | **0.83** | **0.83** |

(b) **Ablation on feature subset ratio $K/d$.**

| Dataset | AUC $\uparrow$ | | |
|---|---|---|---|
| | $K/d = 0.25$ | $K/d = 0.50$ | $K/d = 0.75$ |
| lawschool | 0.86 $\pm$ 0.0026 | 0.88 $\pm$ 0.0027 | 0.88 $\pm$ 0.0024 |
| electricity | 0.79 $\pm$ 0.0070 | 0.77 $\pm$ 0.0072 | 0.73 $\pm$ 0.0070 |
| fried | 0.86 $\pm$ 0.0024 | 0.91 $\pm$ 0.0015 | 0.89 $\pm$ 0.0007 |
| 2dplanes | 0.82 $\pm$ 0.0015 | 0.87 $\pm$ 0.0015 | 0.88 $\pm$ 0.0014 |
| creditcard | 0.73 $\pm$ 0.0029 | 0.72 $\pm$ 0.0028 | 0.71 $\pm$ 0.0028 |
| pol | 0.66 $\pm$ 0.0031 | 0.82 $\pm$ 0.0014 | 0.82 $\pm$ 0.0014 |
| MiniBooNE | 0.78 $\pm$ 0.0076 | 0.77 $\pm$ 0.0058 | 0.77 $\pm$ 0.0049 |
| jannis | 0.84 $\pm$ 0.0035 | 0.83 $\pm$ 0.0042 | 0.82 $\pm$ 0.0043 |
| nomao | 0.79 $\pm$ 0.0019 | 0.79 $\pm$ 0.0021 | 0.78 $\pm$ 0.0021 |
| vehicle_sensIT | 0.81 $\pm$ 0.0014 | 0.81 $\pm$ 0.0014 | 0.80 $\pm$ 0.0015 |
| gas_drift | 0.88 $\pm$ 0.0009 | 0.86 $\pm$ 0.0010 | 0.86 $\pm$ 0.0009 |
| musk | 0.89 $\pm$ 0.0008 | 0.88 $\pm$ 0.0008 | 0.88 $\pm$ 0.0008 |
| Average | 0.81 | **0.83** | 0.82 |

valuation results, with more complex models generally yielding better performance. Nonetheless, all variations of 2D-OOB outperform 2D-KNN, highlighting the clear advantages of the 2D-OOB approach.

**Number of weak learners**    Increasing the number of weak learners allows for a greater number of data-feature subset pairs to be explored, potentially leading to more accurate estimates. However, as shown in Table 3a, when we increase the number of base learners from 500 to 3000, the detection AUC for each dataset remains relatively unchanged, suggesting convergence of the estimation beyond a certain threshold. Typically, 1000 base learners are sufficient to achieve an equitable joint valuation.

**Feature subset ratio $K/d$**    The feature subset ratio $K/d$ refers to the fraction of the total number of features $d$ that are randomly selected to build each weak learner, where $K$ is the number of selected features. In previous experiments, we used a fixed ratio of $0.50$ (unless otherwise specified). To further investigate the impact of this ratio, we now test two additional values: $0.25$ and $0.75$. The results in Table 3b suggest that in general, the joint valuation capacity of our method is robust to the choice of feature subset ratio.

Apart from the experiments discussed above, we showcase that marginalization of 2D-OOB can either match or surpass state-of-the-art data valuation methods on standard benchmarks in Appendix D.

# 5 Related work

**Data contribution estimation**    In addition to the marginal contribution-based methods discussed in Section 2, many other approaches are emerging in the area of data valuation. Just et al. [19] develop a non-conventional class-wise Wasserstein distance between the training and validation sets and use the gradient information to evaluate each data point, an approach that has also been applied to data selection [20]. Wu et al. [51] extend data valuation to deep neural networks, introducing a training-free data valuation framework based on neural tangent kernel theory. Yoon et al. [52] leverage reinforcement learning techniques to automatically learn data valuation scores by training a regression model. However, all these data valuation methods do not assign importance scores to cells, whereas our method provides additional insights into how individual cells contribute to the data valuations.

**Feature attribution**    Feature attribution is a pivotal research domain in explainable machine learning that primarily aims to provide insights into how individual features influence model predictions. Various effective methods have been proposed, including SHAP-based explanation [33, 34, 27, 8, 7], counterfactual explanation [47, 18, 39, 35, 36], and backpropagation-based explanation [1, 2, 45, 44, 53]. Among these methods, the SHAP-based explanation stands out as the most widely adopted approach, utilizing cooperative game theory principles to compute the Shapley value [41]. While feature attribution offers a potential method to attribute data valuation scores across individual cells, our empirical experiments in Appendix B.1 reveal that this two-stage scheme falls short in efficacy compared to our proposed joint valuation paradigm, which integrates data valuation and feature attribution in a simultaneous process.

# 6 Conclusion

We propose `2D-OOB`, an efficient joint valuation framework that assigns a score to each cell in a dataset, thereby facilitating finer attribution of data contributions and enabling a deeper understanding of the dataset. Through comprehensive experiments, we show that `2D-OOB` is computationally efficient and competitive over state-of-the-art methods in multiple joint valuation use cases.

**Discussion**    We emphasize that the primary objective of the joint valuation framework is to evaluate the quality of cells within the dataset, rather than to optimize model performance. The model used serves as a proxy for this evaluation, and it is important to note that a high-performing machine learning method does not necessarily ensure a justified valuation framework.

**Limitation and future work**    While our study primarily explores random forest models applied to tabular datasets and simple image datasets, the potential application of neural network models within the `2D-OOB` framework for more complex vision and language tasks presents a promising avenue for future investigation. For instance, in text datasets, tokens or words can be treated as cells. `2D-OOB` can be easily integrated into any bagging training scheme that uses language models.

Overall, we believe that our work will inspire further exploration in the field of joint valuation, with the broader goal of improving the transparency and interpretability of machine learning, as well as developing an equitable incentive mechanism for data sharing.

## Acknowledgement

We acknowledge computing resources from Columbia University's Shared Research Computing Facility project, which is supported by NIH Research Facility Improvement Grant 1G20RR030893-01, and associated funds from the New York State Empire State Development, Division of Science Technology and Innovation (NYSTAR) Contract C090171, both awarded April 15, 2010.

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

## Supplementary Materials

In the supplementary materials, we provide implementation details, additional experimental results, rigorous formalized proofs and data valuation experiment results. Code repository can be found at `https://github.com/yifansun99/2D-OOB-Joint-Valuation`.

## A Implementation details

### A.1 Datasets

**Tabular datasets**    We use 12 binary classification datasets obtained from OpenML [12]. A summary of all the datasets is provided in Table 4. These datasets are used in Section 4.1, 4.2, 4.4, and Appendix D.

For each dataset, we first employ a standard normalization procedure, where each feature is normalized to have zero mean and unit standard deviation. After preprocessing, we randomly partition a subset of the data into two non-overlapping sets: a training dataset and a test dataset, which consists of 1000 and 3000 samples respectively. The training dataset is used to obtain the joint (or marginal) valuation for each cell (or data point). The test dataset is exclusively used for the cell fixation (or point removal) experiment when evaluating the test accuracy. Note that for methods that need a validation dataset such as `KNNShapley` and `DataShapley`, we additionally sample a separate validation dataset (disjoint from training dataset and test dataset) to evaluate the utility function. The size of the validation dataset is set to 10% of the training sample size.

**Image datasets**    We create datasets by pairing CIFAR-10 classes, each pair consisting of a target attack class and a source class. The training and test dataset comprises 1000 and 2000 samples, respectively. The size of the validation dataset is set to 10% of the training sample size. To manage the computational challenges posed by the baseline method, we employ the super-pixel technique to transform the (32,32,3) image into a 256-dimensional vector. Specifically, we first average the pixel values across the three channels for each pixel. Then, we partition these transformed images into equal-sized $2 \times 2$ grids. Average pooling is applied within each grid to reduce pixel values to a single cell value, which is then arranged into a flattened input vector. A cell is annotated as poisoned if at least 25% of its corresponding grid area contains the trigger.

### A.2 Implementation details for different methods

`2D-OOB`    `2D-OOB` involves fitting a *subset* random forest model with $B = 1000$ decision trees based on the package "scikit-learn". When constructing each decision tree, we fix the feature subset size ratio as 0.5 (unless otherwise specified). For Section 4.3 and Appendix D, we simply adopt $T(y_i, \hat{f}(x_{i,S_b})) = \mathbb{1}(y_i = \hat{f}(x_{i,S_b}))$. For Section 4.1 and 4.2, we further calculate the normalized negative $L_2$ distance between covariates and the class-specific mean in the bootstrap dataset, denoted as $d_{norm}(x_i, y_i)$. Then we use $T(y_i, \hat{f}(x_{i,S_b})) = \mathbb{1}(y_i = \hat{f}(x_{i,S_b})) + d_{norm}(x_i, y_i)$.

`2D-KNN`    `2D-KNN` employs KNN as a surrogate model to approximate `2D-Shapley`. We set the number of nearest neighbors as 10 and the number of permutations as 1000. The hyperparameters are selected based on convergence behavior, and the run time is measured until the values converge.

### A.3 Implementation details for cell-level outlier generation

Following Du et al. [9] and Liu et al. [32], we replace a given cell with an outlier value. Here, the outlier value is randomly generated from the two-sided "tails" of the Gaussian distribution fitted to the column's mean and standard deviation, where the probability of the two-sided tail area is set to be 1%. In total, 4% (20% × 20%) of the cells are replaced with the corresponding outlier values.

### A.4 Implementation details for backdoor trigger generation

Following the prior work [14, 31], we generate the BadNets square and the Trojan square trigger. For BadNets, we adopt the implementation in Nicolae et al. [37]. For Trojan Attack, we use a

pretrained ResNet-18 model on the CIFAR-10 dataset and employ the implementation in Pang et al. [38]. For each attack, we evaluate its effectiveness by training a decision tree model on the poisoned dataset. The accuracy on a clean test set remains nearly unchanged compared to a model trained on an uncontaminated training set, while the attack success rate on a held-out poisoned test set is guaranteed to exceed 75%.

Table 4: **A summary of all the datasets used in 4.1, 4.2, and Appendix D.** These datasets have been commonly used in previous literature [13, 25, 26]

| Name | Total sample size | Input dimension | Majority class proportion | OpenML ID |
|---|---|---|---|---|
| lawschool | 20800 | 6 | 0.679 | 43890 |
| electricity | 38474 | 6 | 0.500 | 44080 |
| fried | 40768 | 10 | 0.502 | 901 |
| 2dplanes | 40768 | 10 | 0.501 | 727 |
| creditcard | 30000 | 23 | 0.779 | 42477 |
| pol | 15000 | 48 | 0.664 | 722 |
| MiniBooNE | 72998 | 50 | 0.500 | 43974 |
| jannis | 57580 | 54 | 0.500 | 43977 |
| nomao | 34465 | 89 | 0.715 | 1486 |
| vehicle_sensIT | 98528 | 100 | 0.500 | 357 |
| gas_drift | 5935 | 128 | 0.507 | 1476 |
| musk | 6598 | 166 | 0.846 | 1116 |

# B    Additional experimental results

## B.1    Additional results for cell-level outlier detection

**Additional results on multi-class classification datasets**    We conducted cell-level outlier detection experiments (as described in Section 4.1) on three multi-class classification datasets from the UCI Machine Learning repository [21]. As shown in the Table 5, `2D-OOB` displays superior detection performance and efficiency.

Table 5: **Cell-level outlier detection results on multi-class classification datasets.** The average and standard error of the AUC and run-time (in seconds) based on 30 independent experiments are denoted by "average $\pm$ standard error".

| Dataset | AUC $\uparrow$ | | Run-time $\downarrow$ | |
|---|---|---|---|---|
| | 2D-OOB (ours) | 2D-KNN | 2D-OOB (ours) | 2D-KNN |
| Covertype | **0.81$\pm$0.0156** | 0.63$\pm$0.0183 | **3.98$\pm$0.5774** | 962.34$\pm$1.3383 |
| Dry Bean | **0.88$\pm$0.0059** | 0.85$\pm$0.0192 | **3.31$\pm$0.4586** | 347.80$\pm$2.0212 |
| Wine Quality | **0.86$\pm$0.0178** | 0.57$\pm$0.0252 | **2.90$\pm$0.1240** | 269.14$\pm$1.1825 |

**Additional baseline: two-stage attribution**    Once we obtain the data valuation scores, an alternative solution approach to determining cell-level attributions involves leveraging feature attribution methods such as SHAP [33]. We explore an additional baseline method building upon this idea: initially, `Data-OOB` (or any other data valuation method) is computed for the $i$-th data point, denoted as $dv_i$. Subsequently, `TreeSHAP` [34] is fitted, using $dv_i$ as the target and the concatenation of $x_i$ and $y_i$ (denoted as $x_i \oplus y_i$) as the predictor. The derived local feature attributions are then interpreted as joint valuation results. We refer to this method as "two-stage attribution".

Table 6 indicates that `2D-OOB` substantially outperforms its two-stage counterpart. We hypothesize that the superiority of our method stems from integrating data valuation and feature attribution into a cohesive framework. Conversely, the two-stage method treats data valuation and feature attribution as separate processes, potentially resulting in sub-optimal outcomes. Furthermore, due to the computational complexity of `TreeSHAP`, the two-stage approach is notably slower compared to our method.

Table 6: **Cell-level outlier detection results (AUC) of** `2D-OOB` **and the two-stage attribution.** Our method shows a better performance than the alternative method by a significant performance margin.

| Dataset | AUC ↑ | |
| --- | --- | --- |
| | 2D-OOB (ours) | Two-stage attribution |
| lawschool | **0.88± 0.0027** | 0.83± 0.0064 |
| electricity | **0.77± 0.0072** | 0.64± 0.0093 |
| fried | **0.91± 0.0015** | 0.82± 0.0068 |
| 2dplanes | **0.87± 0.0015** | 0.80± 0.0058 |
| creditcard | **0.72± 0.0028** | 0.67± 0.0051 |
| pol | **0.82± 0.0014** | 0.78± 0.0042 |
| MiniBooNE | **0.77± 0.0058** | 0.70± 0.0041 |
| jannis | **0.83± 0.0042** | 0.62± 0.0043 |
| nomao | **0.79± 0.0021** | 0.71± 0.0041 |
| vehicle_sensIT | **0.81± 0.0014** | 0.64± 0.0033 |
| gas_drift | **0.86± 0.0010** | 0.73± 0.0143 |
| musk | **0.88± 0.0008** | 0.68± 0.0028 |
| Average | **0.83** | 0.72 |

Table 7: **Cell-level outlier detection results (AUC) of different joint valuation methods when the row outlier ratio and column outlier ratio are both** $50\%$**.** Our method consistently outperforms `2D-KNN` even in the presence of significant noise.

| Dataset | AUC ↑ | |
| --- | --- | --- |
| | 2D-OOB (ours) | 2D-KNN |
| lawschool | **0.75± 0.0084** | 0.60± 0.0144 |
| electricity | **0.64± 0.0155** | 0.60± 0.0106 |
| fried | **0.74± 0.0087** | 0.54± 0.0027 |
| 2dplanes | **0.74± 0.0063** | 0.55± 0.0033 |
| creditcard | **0.63± 0.0055** | 0.61± 0.0053 |
| pol | **0.69± 0.0069** | 0.60± 0.0042 |
| MiniBooNE | **0.67± 0.0128** | 0.60± 0.0048 |
| jannis | **0.70± 0.0113** | 0.53± 0.0014 |
| nomao | **0.70± 0.0088** | 0.58± 0.0052 |
| vehicle_sensIT | **0.70± 0.0075** | 0.55± 0.0031 |
| gas_drift | **0.73± 0.0077** | 0.65± 0.0114 |
| musk | **0.77± 0.0063** | 0.64± 0.0038 |
| Average | **0.71** | 0.59 |

**A noisy setting with more outlier cells**  We consider a more challenging scenario with increased outlier levels, where both the row outlier ratio and column outlier ratio increase from $20\%$ (as in Section 4.1) to $50\%$. Consequently, this leads to $25\%$ ($50\% \times 50\%$) of the cells being replaced with outlier values. We follow the same outlier generation procedure outlined in Appendix A.3. The findings, presented in Table 7, demonstrate that our method maintains a significantly superior performance over `2D-KNN`, even under such a noisy setting.

### B.2   Additional results for cell fixation experiment

Figure 6 presents the results for the cell fixation experiment on 6 additional datasets. `2D-OOB` excels in precisely detecting and correcting relevant cell outliers.

**The scenario without ground-truth knowledge**  Following Liu et al. [32], we examine a situation where external information on the ground-truth annotations of outlier cells is not accessible. In this scenario, we address these outliers by substituting them with the average of other cells in the same feature column. This procedure starts by addressing cells with the lowest valuations, based on the hypothesis that correcting these cells is likely to maintain or potentially improve the model's performance. As depicted in Figure 7, `2D-OOB` conforms to this expected trend, demonstrating the

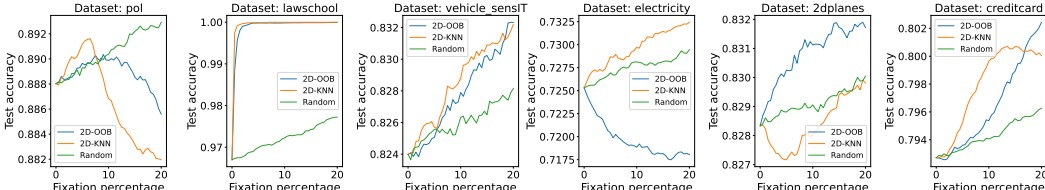

Figure 6: **Cell fixation experiment results (test accuracy curves) for** `2D-OOB`, `2D-KNN` **and a random baseline.** We replace cell values with ground-truth values from the cells with the lowest valuation to the highest valuation. The results from 6 additional datasets are displayed. We conduct 30 independent trials and report the average results. A higher curve indicates better performance. `2D-OOB` sets itself apart by its remarkable precision in detecting and rectifying relevant cell outliers.

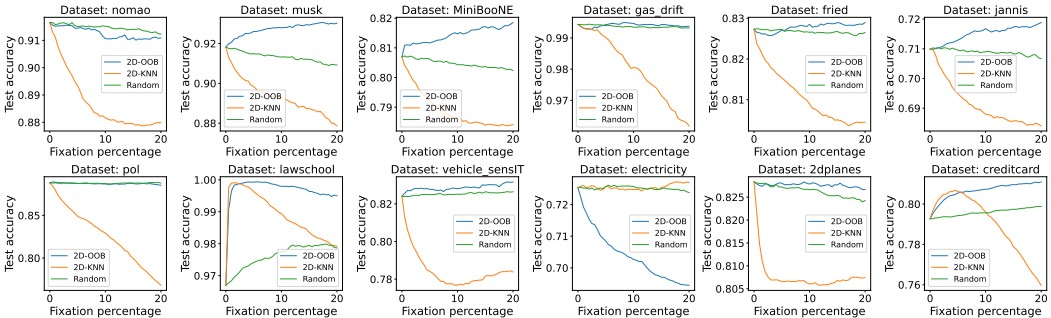

Figure 7: **Cell fixation experiment (without ground-truth knowledge) results (test accuracy curves) for** `2D-OOB`, `2D-KNN` **and a random baseline.** We replace cell values with column mean imputations from cells with the lowest value to the highest value. We report the average results of 30 independent trials from 12 datasets. A higher curve indicates better performance.

effectiveness of our method in joint valuation. Conversely, `2D-KNN` fails to show similar performance improvements.

### B.3 Additional results for backdoor trigger experiment

We provide additional qualitative examples of the backdoor trigger detection experiment in Figure 8.

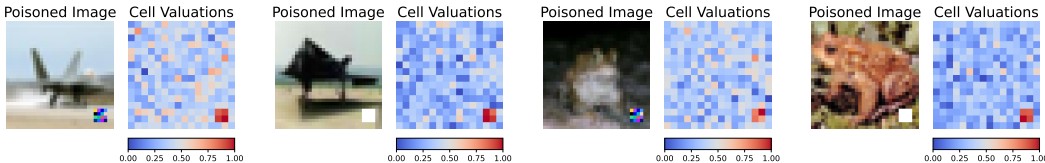

Figure 8: **Qualitative results on more datasets for the backdoor trigger detection experiment.** The first two images are from the class "airplane" but have been relabeled as "automobile", while the latter two images are from the class "frog" and have been relabeled as "horse".

## C Proof of Proposition 3.1

*Proof.* For simplicity, we denote $\phi_i^{\mathrm{OOB}}(S)$ as $\phi_i(S)$ and $\psi_{ij}^{\mathrm{2D-OOB}}$ as $\psi_{ij}$ in the proof. Let $\mathcal{S} := \{S \subseteq [d]\}$ represent the set of all feature subsets, where $S$ is a feature subset. We denote the cardinality of $\mathcal{S}$ as $L := |\mathcal{S}| = 2^d$. Let $\boldsymbol{\gamma}_b$ be a weight vector $\boldsymbol{\gamma}_b := (\gamma_{b1}, \ldots, \gamma_{bL})$ for all $b \in [B]$, where $\gamma_{bl} \in \{0, 1\}$ and $\gamma_{bl} = 1$ indicates the $l$-th subset is used in the $b$-th weak learner. With $\{\mathbf{w}_b, \boldsymbol{\gamma}_b, \hat{f}_b\}_{b=1}^B$, we can denote the $i$-th `Data-OOB` on the $l$-th feature subset $S_l$ as

$$\phi_i(S_l) = \frac{\sum_{b=1}^B \mathbb{1}(w_{bi} = 0)\mathbb{1}(\gamma_{bl} = 1)T(y_i, \hat{f}_b(x_{i,S_l}))}{\sum_{b=1}^B \mathbb{1}(w_{bi} = 0)\mathbb{1}(\gamma_{bl} = 1)}.$$

With slight abuse of notation, the formulation of 2D-OOB in (4) can be expressed as follows.

$$
\begin{aligned}
\psi_{ij} &= \frac{\sum_{l=1}^{L} \sum_{b=1}^{B} \mathbb{1}(w_{bi}=0)\mathbb{1}(\gamma_{bl}=1)\mathbb{1}(j \in S_l)T(y_i, \hat{f}_b(x_{i,S_l}))}{\sum_{l=1}^{L} \sum_{b=1}^{B} \mathbb{1}(w_{bi}=0)\mathbb{1}(\gamma_{bl}=1)\mathbb{1}(j \in S_l)} \\
&= \sum_{l=1}^{L} \mathbb{1}(j \in S_l) \frac{\sum_{b=1}^{B} \mathbb{1}(w_{bi}=0)\mathbb{1}(\gamma_{bl}=1)T(y_i, \hat{f}_b(x_{i,S_l}))}{\sum_{l=1}^{L} \sum_{b=1}^{B} \mathbb{1}(w_{bi}=0)\mathbb{1}(\gamma_{bl}=1)\mathbb{1}(j \in S_l)} \\
&= \sum_{l=1}^{L} \mathbb{1}(j \in S_l) \frac{\sum_{b=1}^{B} \mathbb{1}(w_{bi}=0)\mathbb{1}(\gamma_{bl}=1)}{\sum_{l=1}^{L} \sum_{b=1}^{B} \mathbb{1}(w_{bi}=0)\mathbb{1}(\gamma_{bl}=1)\mathbb{1}(j \in S_l)} \frac{\sum_{b=1}^{B} \mathbb{1}(w_{bi}=0)\mathbb{1}(\gamma_{bl}=1)T(y_i, \hat{f}_b(x_{i,S_l}))}{\sum_{b=1}^{B} \mathbb{1}(w_{bi}=0)\mathbb{1}(\gamma_{bl}=1)} \\
&= \sum_{l=1}^{L} \alpha_{i,j,l}\phi_i(S_l),
\end{aligned}
$$

where $\alpha_{i,j,l} \propto \mathbb{1}(j \in S_l)\sum_{b=1}^{B}\mathbb{1}(w_{bi}=0)\mathbb{1}(\gamma_{bl}=1), \forall i \in [n], j \in [d], l \in [L]$ and $\sum_{l=1}^{L}\alpha_{i,j,l} = 1$. Define $P_i(S_l | j \in S_l, \{w_{bi}\}_{b=1}^{B}) = \alpha_{i,j,l}$, which specifies an empirical distribution of the feature subset $S$, conditioned on $j \in S$ and the bootstrap sampling process. Here, $\mathbb{1}(j \in S_l)$ indicates that the distribution is conditioned on the inclusion of the $j$-th feature within the feature subset $S_l$. $w_{bi}$ indicates whether the $i$-th sample is out-of-bag in the $b$-th bootstrap, and $\gamma_{bl}$ indicates whether the $l$-th feature subset is selected in the $b$-th weak learner. Thus, the point mass is determined by the sampling process, leading to:

$$
\psi_{ij} = \mathbb{E}_{\hat{F}_S}[\phi_i(S) \mid j \in S].
$$

$\square$

## D  Data valuation experiment

In this section, we show that 2D-OOB-data, the marginalization of 2D-OOB, offers an effective approach to data valuation. This serves as the basis of our enhanced performance in joint valuation.

**Marginalization**  2D-OOB aims to attribute data contribution through cells. Consequently, by summing up 2D-OOB over all columns, we can derive data contribution values. For $i \in [n]$, we define the 2D-OOB-data $\psi_i^{data}$ as follows.

$$
\psi_i^{data} := \frac{1}{d}\sum_{j=1}^{d} \psi_{ij}^{\text{2D}-\text{OOB}}. \tag{5}
$$

Based on discussions in Section 3.2, the marginalizations also connect with Data-OOB:

**Proposition D.1.** *For all $i \in [n]$, the marginalizations $\psi_i^{data}$ can be expressed as follows.*

$$
\psi_i^{data} = \mathbb{E}_{\hat{F}_S}[\phi_i^{OOB}(S)],
$$

*where the notations follow the same definitions as Proposition 3.1.*

*Proof.* Based on the definition of 2D-OOB-Data, for $i \in [n]$,

$$
\begin{aligned}
\psi_i^{data} := \frac{1}{d}\sum_{j=1}^{d}\psi_{ij}^{\text{2D}-\text{OOB}} &= \frac{1}{d}\sum_{j=1}^{d}\sum_{l=1}^{L}\alpha_{i,j,l}\phi_i^{\text{OOB}}(S_l) \\
&= \sum_{l=1}^{L}(\frac{1}{d}\sum_{j=1}^{d}\alpha_{i,j,l})\phi_i^{\text{OOB}}(S_l),
\end{aligned}
$$

where $\alpha_{i,j,l}$ is defined in Appendix C. We have $\sum_{l=1}^{L}(\frac{1}{d}\sum_{j=1}^{d}\alpha_{i,j,l}) = \frac{1}{d}\sum_{j=1}^{d}\sum_{l=1}^{L}\alpha_{i,j,l} = 1$. Denote $\mathbb{P}_i(S_l|\{w_{bi}\}_{b=1}^{B}) = \frac{1}{d}\sum_{j=1}^{d}\alpha_{i,j,l}$, which induces the empirical expectation of Data-OOB with respect to $S_l$. $\square$

Proposition D.1 indicates `2D-OOB-data` $\psi_i^{data}$ can be expressed as the average `Data-OOB` value for the $i$-th data point. As a result, `2D-OOB-data` is expected to inherit the advanced ability of `Data-OOB` in terms of data valuation, as will be empirically examined next.

**Experimental setting**   Following the standard protocol in Kwon and Zou [25, 26] and Jiang et al. [17], we randomly select 10% of the data points and flip its label to the other class. For joint valuation methods, we calculate the valuation of each cell and perform the marginalization over features to obtain the data valuation scores. Mislabeled data detection and data removal experiments are examined based on this setting. For the baseline methods, we further incorporate several state-of-the-art data valuation methods including `DataShapley` [13], `KNNShapley` [16], `DataBanzhaf` [50], `LAVA` [19], and `Data-OOB` [26]. Implementation details are listed below. To guarantee a fair comparison, we also employ the decision tree as the base model in `DataShapley` and `DataBanzhaf`. We adopt the same 12 datasets as outlined in Section 4.1.

`Data-OOB`   `Data-OOB` involves fitting a random forest model without feature subset sampling, consisting of 1000 decision trees.

`DataShapley`   We use a Monte Carlo-based algorithm. The Gelman-Rubin statistics is computed to determine the termination criteria of the algorithm. Following Jiang et al. [17], We adopt the threshold to be 1.05.

`KNNShapley`   We set the number of nearest neighbors to be 10% of the sample size following Jia et al. [16].

`LAVA`   We calculate the class-wise Wasserstein distance following Just et al. [19]. The "OTDD" framework is adopted to complete the optimal transport calculation.

`DataBanzhaf`   We adopt the implementation from Jiang et al. [17]. We set "the number of models to train" as 1000.

## D.1   Mislabeled data detection

We calculate the precision-recall curve by comparing the actual annotations, which denote whether data points are mislabeled, against the data valuation scores computed by different methods. Mislabeled data typically have a detrimental impact on model performance. Therefore, data points that receive a lower valuation score are considered to have a higher chance of being mislabeled. We then determine AUCPR (the AUC of the precision-recall curve) as a quantitative metric to assess the detection efficacy.

As shown in Table 8, `2D-OOB-data` consistently outperforms `2D-KNN-data` across all datasets, suggesting its superior ability to detect mislabeled data points. It is worth noting that `2D-OOB-data`'s results are on par with `Data-OOB`, while significantly exceeding the performance of other data valuation methods. These results are in line with our theoretical analysis regarding the resemblance between `Data-OOB` and `2D-OOB-data`. However, it is important to highlight that applying `Data-OOB` to the joint tasks is not feasible as mentioned earlier, underscoring the necessity for the development of `2D-OOB`.

## D.2   Point removal experiment

Removing low-quality data points has the potential to enhance model performance. Based on this idea, we employ the point removal experiment, a widely used benchmark in data valuation [26, 13, 25]. According to the calculated data valuation scores, we progressively remove data points from the dataset in *ascending* order. Specifically, we begin by removing the data points with the lowest data valuations. Each time we remove a datum, we fit a logistic model and use the held-out test set consisting of 3000 instances to evaluate the model performance. The expected behavior is that the model performance will improve initially as the detrimental data points are gradually eliminated from the training process. Removing an excessive number of data points may result in a drastically altered dataset. Consequently, we opt to remove the bottom 20% data points.

Table 8: **Point-level mislabeled data detection results.** AUCPR of different data valuation and (marginalized) joint valuation methods. The average and standard error of the AUCPR based on 30 independent experiments are denoted by "average ± standard error". Bold numbers denote the best method, for data valuation and joint valuation respectively. The AUCPR value for the `Random` method consistently remains at 0.5 across all datasets. `2D-OOB-data` exhibits performance comparable to `Data-OOB`, while significantly surpassing `2D-KNN-data` (the marginalization of 2D-KNN) and all other data valuation methods.

| Dataset | Data Valuation | | | | | Joint Valuation (Marginalized) | |
| | KNNShapley | LAVA | DataBanzhaf | DataShapley | Data-OOB | 2D-KNN-data | 2D-OOB-data (ours) |
|---|---|---|---|---|---|---|---|
| lawschool | 0.66 ± 0.013 | 0.13 ± 0.003 | 0.46 ± 0.008 | 0.88 ± 0.007 | **1.00 ± 0.000** | 0.46 ± 0.011 | **0.99 ± 0.002** |
| electricity | 0.22 ± 0.008 | 0.11 ± 0.002 | 0.18 ± 0.005 | 0.26 ± 0.007 | **0.44 ± 0.007** | 0.20 ± 0.006 | **0.39 ± 0.007** |
| fried | 0.40 ± 0.014 | 0.11 ± 0.002 | 0.22 ± 0.007 | 0.35 ± 0.009 | **0.76 ± 0.007** | 0.34 ± 0.010 | **0.73 ± 0.008** |
| 2dplanes | 0.46 ± 0.016 | 0.12 ± 0.002 | 0.32 ± 0.007 | 0.54 ± 0.009 | **0.78 ± 0.008** | 0.44 ± 0.011 | **0.68 ± 0.010** |
| creditcard | 0.37 ± 0.007 | 0.11 ± 0.003 | 0.16 ± 0.004 | 0.28 ± 0.006 | **0.40 ± 0.007** | 0.20 ± 0.005 | **0.40 ± 0.007** |
| pol | 0.19 ± 0.017 | 0.11 ± 0.002 | 0.37 ± 0.010 | 0.58 ± 0.012 | **0.93 ± 0.004** | 0.29 ± 0.018 | **0.87 ± 0.005** |
| MiniBooNE | 0.41 ± 0.013 | 0.13 ± 0.006 | 0.23 ± 0.007 | 0.41 ± 0.010 | **0.78 ± 0.007** | 0.36 ± 0.008 | **0.78 ± 0.007** |
| jannis | 0.20 ± 0.007 | 0.11 ± 0.002 | 0.14 ± 0.003 | 0.17 ± 0.005 | **0.38 ± 0.010** | 0.19 ± 0.006 | **0.37 ± 0.010** |
| nomao | 0.61 ± 0.012 | 0.14 ± 0.003 | 0.33 ± 0.010 | 0.58 ± 0.009 | **0.87 ± 0.006** | 0.33 ± 0.011 | **0.88 ± 0.005** |
| vehicle_sensIT | 0.22 ± 0.009 | 0.11 ± 0.002 | 0.21 ± 0.007 | 0.33 ± 0.011 | **0.56 ± 0.010** | 0.14 ± 0.005 | **0.56 ± 0.010** |
| gas_drift | 0.87 ± 0.013 | 0.16 ± 0.006 | 0.42 ± 0.009 | 0.75 ± 0.008 | **0.98 ± 0.002** | 0.88 ± 0.006 | **0.98 ± 0.002** |
| musk | 0.33 ± 0.010 | 0.11 ± 0.003 | 0.31 ± 0.007 | 0.47 ± 0.012 | **0.85 ± 0.005** | 0.21 ± 0.008 | **0.85 ± 0.005** |
| Average | 0.41 | 0.12 | 0.28 | 0.47 | **0.73** | 0.34 | **0.71** |

Test accuracy curves throughout the data removal process are shown for 12 datasets (Figure 9). A higher curve signifies better performance in terms of data valuation. Overall, `2D-OOB-data` demonstrates similar performance to `Data-OOB`, while significantly outperforming all other data valuation methods and the random baseline. When a few data points with poor quality are removed, the test performance of `2D-OOB-data` exhibits an evident increase. However, such a positive trend does not apply to other popular data valuation methods including `DataShapley` and `LAVA`. These findings highlight the potential of `2D-OOB-data` in selecting a subset of critical data points that can maintain model performance when the dataset is pruned.

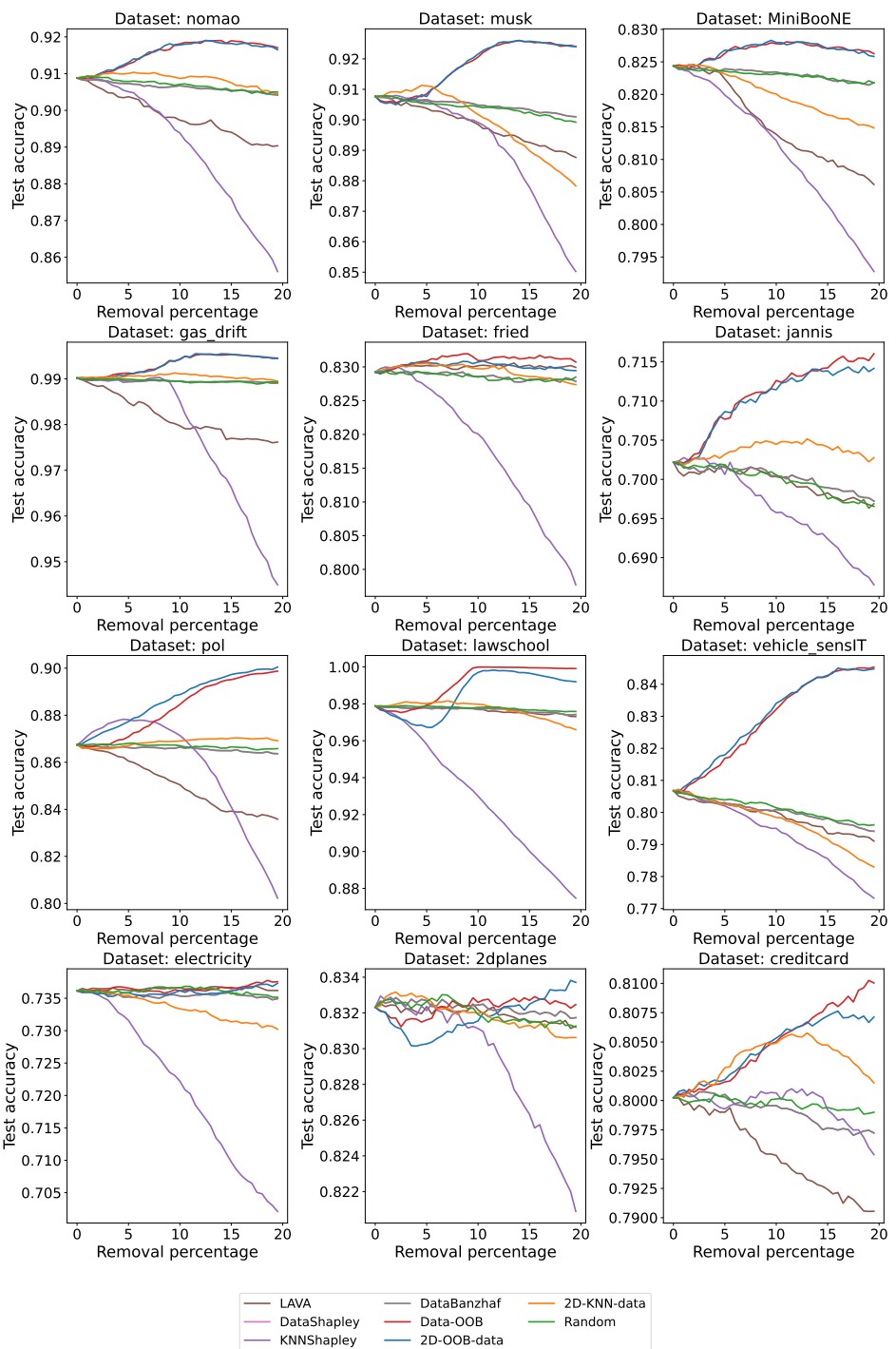

Figure 9: **Point removal experiment results (test accuracy curves) of** 7 **data valuation methods –** `2D-OOB-data`, `2D-KNN-data`, `Data-OOB`, `LAVA`, `DataBanzhaf`, `DataShapley`, `KNNShapley` **and a random baseline.** We remove data points from the lowest valuation to the highest valuation. The results from 6 binary classification datasets are displayed. For each dataset, we conduct 30 independent trials and report the average results. A higher curve indicates better performance. `2D-OOB-data` demonstrates superior ability in finding a set of helpful data points.

