# OpenReview forum: "2D-OOB: Attributing Data Contribution Through Joint Valuation Framework"
_NeurIPS.cc/2024/Conference — NeurIPS 2024 poster_

### Official Review · Reviewer_ECZV · 2024-07-12

**Soundness:** 3
**Presentation:** 3
**Contribution:** 3
**Rating:** 6
**Confidence:** 4

**Summary:**

This paper studies data attribution. Unlike the previous works that only quantifies the contribution of each datum, this work also considers the attribution of the cells within every datum. The proposed the joint valuation framework, 2D-OOB, can accurately identify cell outliers, as well as the poisoning attacks. The experiments verify the authors claims and show the efficiency of the proposed method.

**Strengths:**

1. The paper is well-written and easy to follow.
2. The proposed method has been theoretically generalized and connected to the existing works.
3. The experiments are comprehensive and show the efficiency of the proposed method.

**Weaknesses:**

1. In section 4.3, the experiments did not mention that the classification label is altered or not after poisoning (e.g, from correct to incorrect).
2. The paper does not state clearly about the differences of the application on image, comparing with the other model attribution methods? They both assign a value to each feature/pixel/cell in the input data. Why would the proposed method be better in this scenario?
3. How about the scalability of the proposed method? The experiments are either conducted on tabular data or small images. The large images or high-dimensional data are not mentioned.
4. What is the distance regularization term mentioned in L168 in the utility function $T$?
5. The size of CIFAR-10 is 32x32, but the cell valuation displayed in Figure 5 is 16x16. Why is that?

**Questions:**

See weaknesses

---

> ### Author Rebuttal · Authors · 2024-08-06
>
> ### **Weakness-1: Section 4.3 Clarification**
>
> **Re.:**  In line 216 on page 7, we explicitly stated that the poisoned samples are relabeled from the source class to the target class.
>
> ----
>
> ### **Weakness-2: Difference between 2D-OOB and image feature attribution methods**
>
> **Re.:** Thank you for pointing this out. As the reviewer mentioned, both feature attribution methods and the joint valuation method we studied in this paper assign a score to each cell. However, we would like to clarify that they serve distinct purposes and are conceptually different.
>
> Feature attribution methods measure the impact of a particular data point’s features on its model prediction. In the case of image classification, these methods explain which parts of an image are most influential for its model's prediction.
> In contrast, the joint valuation method measures the impact of each cell in the training dataset on the model training process. Hence, it explains which parts of training images are most influential or potentially harmful to the model's performance.
>
> ----
>
> ### **Weakness-3: Scalability**
>
> **Re.:** We have included the additional experimental results on the scalability of 2D-OOB in the general response.
>
> ----
>
> ### **Weakness-4:Regularization term**
>
> **Re.:** We calculate the normalized (min-max normalization) negative L2 distance between covariates and the class-specific mean in the bootstrap dataset, as stated in L439-440. The normalization is performed across those samples within the same class. We will present a more explicit formulation in the revision.
>
> ----
>
> ### **Weakness-5: Super-pixel**
>
> **Re.:**  We employ a standard super-pixel technique to transform the (32,32,3) image into a 256-dimensional vector (This is explicitly stated in line 427 page 13 of our submitted manuscript). Specifically, we partitioned the images into equally sized $2\times2$ grids and used average pooling to reduce the pixel values to a single cell value. The super-pixel technique we employed has been widely used in numerous previous explainable machine learning works [1-4] to highlight key factors in a simplified illustration.
>
> [1] Dardouillet, Pierre, et al. "Explainability of image semantic segmentation through shap values." International Conference on Pattern Recognition. Cham: Springer Nature Switzerland, 2022.
> [2] Vermeire, Tom, et al. "Explainable image classification with evidence counterfactual." Pattern Analysis and Applications 25.2 (2022): 315-335.
> [3] Shah, Sumeet S., and John W. Sheppard. "Evaluating explanations of convolutional neural network image classifications." 2020 International Joint Conference on Neural Networks (IJCNN). IEEE, 2020.
> [4] Jethani, Neil, et al. "Fastshap: Real-time shapley value estimation." International Conference on Learning Representations. 2021.

---

> > ### Comment · Reviewer_ECZV · 2024-08-13
> >
> > Thanks for the authors' response, which has addressed most of my concerns. The authors should include the key points of the discussion into their revised version.

---

> > > ### Comment · Area_Chair_idkm · 2024-08-13
> > >
> > > Dear authors,
> > >
> > > Are you planning to revise the paper accordingly?

---

> > > > ### Author Response · Authors · 2024-08-13
> > > > **Thank you!**
> > > >
> > > > Dear AC and Reviewer ECZV,
> > > >
> > > > We are grateful for your feedback and suggestions. We will make sure to include necessary clarifications, discussion on our method and image feature attribution, as well as scalability results in the revision.

---

### Official Review · Reviewer_nD8c · 2024-07-14

**Soundness:** 3
**Presentation:** 3
**Contribution:** 2
**Rating:** 5
**Confidence:** 3

**Summary:**

The paper proposes 2D-OOB, an out-of-bag estimation framework for jointly detecting both samples as well as their cells that are outliers. In other words, 2D-OOB allows attribution of the value of a data sample to its individual features.  Evaluation shows that the framework is also able to localize backdoor triggers in data poisoning attacks.
In comparison to the 2D-Shapley model, which also performs joint valuation of a sample and its features, the 2D-OOB is a more efficient and model-agnostic formulation and builds on the subset bagging model, where each weak learner is trained on a randomly selected subset of features. In the experiments, decision trees as used as the weak learners and trained using randomly selected features. The framework measures (using Eqn. (4)), the average accuracy of the bag of predictors for a selected sample and its features.

**Strengths:**

* The paper is well written.
* Given the background in moving fro DataShapley to the Data-OOB formulation, the transition from 2D-Shapley to 2D-OOB was intuitive and easy to follow.
* The proposed method can identify data poisoning attack triggers (square patches placed in the original images to elicit misclassification) which could be very useful.

**Weaknesses:**

* The paper is low on novelty. Given, OOB-Data formulation and the 2D-Shapley for joint valuation, the 2D-OOB formulation seems like a natural extension -- interesting, but not high on novelty.

* In Fig. 4, panel B, for BadNets poisoning attack, the detection performance of 2D-OOB and 2D-KNN (proxy for 2D-Shapley) are very similar. It is difficult to assess how 2D-OOB will perform for more stronger poisoning attacks. A more detailed study on this aspect as well as analysis showing that 2D-OOB is more likely to identify such triggers, even when attack takes such a detection mechanism into account, will be a good indicator of the strength of this framework.

* Table 1, shows the average run time is much improved from 2D-KNN, however, most of the datasets used (Appendix A, Table 3) have small input dimensions (only three have >= 100 but have small sample size) and hence it is difficult to assess how the technique would scale.

**Questions:**

* One of the arguments mentioned in the description of Fig. 2 is that 2D-OOB detects a majority of outlier cells by examining a small fraction of the total cells. It would be great if the authors could further explain this statement and which part of the formulation in (4) motivates this argument.

* In Fig. 3 (showing 2D-OOBs ability to identify and prioritize the outliers to fix), the test accuracy for the Jannis dataset is relatively low compared to the others and the performance is also similar to 2D-KNN. Any insights as to why this is happening?

* In Section 4.4, were the experiments also performed by varying B using a 2-layer MLP? What is the overhead of training an out-of-bag estimator versus a single model (maybe an autoencoder) and using that for outlier detection.

* Are there any other stronger baselines against which comparison can be made?

**Limitations:**

Yes.

---

> ### Author Rebuttal · Authors · 2024-08-06
>
> ### **Weakness-1: Novelty**
>
> **Re.:** Thank you for your feedback. We acknowledge that 2D-OOB may initially appear to be a natural extension of existing methods, as it is built on them. However, we believe our contribution is substantial in the field of data valuation, as our proposed method addresses both practical and conceptual challenges of 2D-Shapley and Data-OOB.
>
> Compared to 2D-Shapely, our work significantly improves **computational efficiency**, as demonstrated by the empirical experiments in the submitted paper and the new results in our general response. In addition, we identify **new** use cases of the joint valuation framework, such as attributing data contribution through cell values, correcting erroneous cells, and detecting backdoor triggers, which have not been discussed in 2D-Shapley.
>
> In comparison with Data-OOB, which only provides data-level values, our method is designed to assess cell-level values. This finer granularity in valuation presents a richer interpretation of data values by providing **attribution for data values**. Also, it enables new practical applications.
>
> ----
>
> ### **Weakness-2: Detailed study on different attacks**
>
> **Re.:**  BadNets and Trojan attacks are popular data poisoning techniques that maliciously contaminate the training datasets by injecting samples patched with a pre-defined trigger [1,2]. To expand our analysis, we explored the impact of varying types of triggers in additional experiments. All variations we used in the experiments are motivated by [1,2]. 2D-OOB consistently demonstrates superior detection capabilities over 2D-KNN across all trigger variations. This result suggests that 2D-OOB has a stronger detection capability against non-random outliers in applications compared to 2D-KNN, as KNN algorithms focus on local proximity and can fail to capture global patterns in feature behavior effectively.
>
> Table: Additional experiment results on backdoor trigger detection. Five independent experiments are conducted on each dataset and the average detection AUC across all datasets is presented. The average and standard error of the detection AUC are denoted by “average ± standard error”.
>
> | Trigger         | 2D-OOB (ours) | 2D-KNN |
> |-------------|-------------|-------------|
> | BadNets White Square  |         0.97$\pm$0.01      |  0.90$\pm$0.03 |
> | BadNets Yellow Square |   0.98 $\pm$0.004      |  0.81$\pm$0.004  |
> | BadNets White Triangle |  0.95 $\pm$0.01         |  0.91$\pm$0.02  |
> | Trojan White Square |  0.92 $\pm$0.005       |  0.76$\pm$0.006 |
> | Trojan Black Apple Logo |  0.88$\pm$0.008      |  0.78$\pm$0.04 |
> | Trojan White Apple Logo |  0.87$\pm$0.006      |  0.76$\pm$0.04 |
>
>
> [1] Tianyu Gu, Brendan Dolan-Gavitt, and Siddharth Garg. Badnets: Identifying vulnerabilities in the machine learning model supply chain. arXiv preprint arXiv:1708.06733, 2017.
> [2] Yingqi Liu, Shiqing Ma, Yousra Aafer, Wen-Chuan Lee, Juan Zhai, Weihang Wang, and X. Zhang. 352 Trojaning attack on neural networks. In Network and Distributed System Security Symposium, 2018. URL 353 https://api.semanticscholar.org/CorpusID:31806516.
>
>
> ----
>
> ### **Weakness-3: Scalability**
>
> **Re.:** We have included the additional experimental results on the scalability of 2D-OOB in the overall response above. Please let us know if there are any other aspects you'd like us to discuss further.
>
> ----
>
> ### **Question-1: Fig. 2 statement that 2D-OOB detects a majority of outlier cells by examining a small fraction of the total cells**
>
> **Re.:** 2D-OOB assigns a quality score to each cell, where smaller values indicate poorer quality and a higher likelihood of being outliers. Consequently, by checking cells with low scores, we can effectively scan out most outliers, as shown in Fig 2.
>
> For Equation 4, the underlying intuition is that if a particular feature of a data point (i.e., a cell) consistently results in a high OOB error when used in weak learners, it indicates that the cell is of poor quality and potentially a harmful outlier, thus deserving a low score.
>
> ----
>
> ### **Question-2: Fig. 3 results**
>
> **Re.:** The relatively low test accuracy observed for the Jannis dataset can be attributed to inherent variations in data separability across different datasets. The Jannis dataset may exhibit more complex patterns or higher noise levels. For example, the covariates may not be useful enough to predict their labels. The performance similarity between 2D-OOB and 2D-KNN indicates that both methods might face similar challenges with this dataset.
>
> ----
>
> ### **Question-3: Section 4.4 results**
>
> **Re.:** For different weak learners (including 2-layer MLP), we use a consistent B of 1000, which is large enough to guarantee convergence empirically.
>
> An autoencoder has the potential to identify outliers, while our out-of-bag joint valuation framework can provide **richer** information (i.e., identify both low-quality and **high-quality** cells). Given the differences in the objective of these methods, comparing the overhead of training an out-of-bag estimator versus a single model like an autoencoder might not be directly meaningful.
>
> We hope this addresses your concern. Please let us know if there are any other aspects you'd like us to discuss further.
>
> ----
>
> ### **Question-4: Stronger baselines**
>
> **Re.:** Thank you for the question. Given that the joint valuation problem is still emerging, with the first concept introduced in 2023 [1], there is no established baseline beyond 2D-KNN. The proposed 2D-OOB demonstrates both practical effectiveness and computational efficiency compared to 2D-KNN in many downstream tasks.
>
> [1] Liu, Zhihong, et al. "2D-Shapley: A Framework for Fragmented Data Valuation." ICML. 2023.

---

> > ### Comment · Reviewer_nD8c · 2024-08-14
> >
> > Thank you for the detailed response. I am satisfied with the scalability experiments as well as the results on the attacks. I am increasing my score to 5.

---

> > > ### Author Response · Authors · 2024-08-14
> > > **Thank you**
> > >
> > > We greatly appreciate your constructive review and are glad our new results addressed your concerns. We will incorporate all these new results and discussions in the revision. Thank you for re-evaluating our paper.

---

### Official Review · Reviewer_4K7L · 2024-07-15

**Soundness:** 3
**Presentation:** 3
**Contribution:** 3
**Rating:** 6
**Confidence:** 5

**Summary:**

This paper proposes a joint valuation framework for not only obtaining data value but also attributing a data points value to its individual features (cells). The framework is build on top of data-OOB. The authors compared the proposed 2D-OOB with 2D-KDD on several tasks, including cell-level outlier detection and backdoor trigger detection in data poisoning attacks.

**Strengths:**

(+) This paper targets an important problem that is useful for improving the interpretability of data valuation. The joint valuation provides more fine-grained information. Identifying outliers at the cell level is informative for determining which cell to fix in the next step when improving data quality, especially useful when data acquisition is expensive.

(+) The proposed method naturally integrates the idea of random forest - random feature selection in the construction of weak learners - with data-OOB, offering a new solution for joint evaluation. This approach addresses a relatively underexplored problem, with few existing works in the area. The proposed 2D-OOB method demonstrates greater computational efficiency compared to 2D-KDD.

(+) The improvement in backdoor trigger detection is promising, as cell contamination in this context is targeted and deliberate, unlike artificial random outliers.

(+) The authors provide an ablation study to assess the impact of the selection and number of weak learners, as well as the feature subset ratio, which helps in understanding the stability of the proposed method.

**Weaknesses:**

(-) Despite the strengths mentioned above, the entire framework is built on top of the OOB framework. This brings a main limitation, as the OOB framework requires the training algorithm to be a bagging method with weak learners. In many applications, bagging may not be the best-performing method, and data valuation under this method may not be of interest.

(-) Typically, the number of randomly selected features for each weak learner is an important hyperparameter to fine-tune. What is the impact when the number of random features is small? While an ablation study was provided in Appendix, certain variation was still observed. Should we correspondingly update the bagging size? Some practical guidance would be helpful.

**Questions:**

1. What does fitting a logistic model in line 187 on page 6 mean? Why change from bagging decision trees to a logistic model in this specific cell fixation experiment?

2. It would be helpful to further explain the difference between Trojan and BadNets trigger methods, as it seems that 2D-OOB is much more effective than 2D-KNN under the Trojan square. This would help us understand in which scenario 2D-OOB works better.

3. Regarding the ablation study: I am also curious about the original prediction performance for each task, not just the outlier detection performance.

4. What is the implication of the connection to data-OOB in Proposition 3.1? Are there any insights we can gain after establishing the connection with data-OOB?

---

> ### Author Rebuttal · Authors · 2024-08-06
>
> ### **Weakness-1: the OOB framework requires the training algorithm to be a bagging method with weak learners**
>
> **Re.:** The primary objective of the joint valuation framework is to evaluate the quality of cells rather than to optimize model performance. The model used serves as a proxy for this evaluation, and it is important to note that a high-performing machine learning method does not necessarily ensure a justified valuation framework.
>
> Additionally, we believe the bagging framework can be useful in many domains[1-3], and 2D-OOB can be easily integrated into these applications.
>
> [1] Smaida, Mahmoud, and Serhii Yaroshchak. "Bagging of Convolutional Neural Networks for Diagnostic of Eye Diseases." COLINS. 2020.
> [2] Dong, Yan-Shi, and Ke-Song Han. "A comparison of several ensemble methods for text categorization." IEEE International Conference onServices Computing, 2004.(SCC 2004). Proceedings. 2004. IEEE, 2004.
> [3] Xinqin, L. I., et al. "Application of bagging ensemble classifier based on genetic algorithm in the text classification of railway fault hazards." 2019 2nd International Conference on Artificial Intelligence and Big Data (ICAIBD). IEEE, 2019.
>
> ----
>
> ### **Weakness-2: Impact of number of random features and bagging size**
>
> **Re.:** If the number of random features is **too small**, each weak learner becomes nearly random, which significantly diminishes the joint valuation capacity. This issue is independent of the bagging size, as nearly random weak learners, regardless of their quantity, fail to provide meaningful signals for effective evaluation.
>
> In practice, for low-dimensional datasets, we recommend using a higher feature subset ratio to ensure sufficient capacity of weak learners. For high-dimensional datasets, a lower or moderate feature subset ratio might be more effective in distinguishing the impact of features.
>
> ----
>
> ### **Question-1: Clarification of fitting a logistic model in line 187 on page 6**
>
> **Re.:** The cell fixation experiment is divided into two separate stages. In the first stage (attribution), we use a subset bagging model with decision trees as weak learners to fit the joint valuation framework. In the second stage (evaluation), our objective is to assess whether low-quality cells were effectively removed, rather than to optimize model performance. Logistic regression was chosen because it is a simple yet powerful machine learning model commonly used to test data separability, which is a common practice in this field [1][2].
>
> [1] Jiang, Kevin, et al. "Opendataval: a unified benchmark for data valuation." NeurIPS. 2023.
> [2] Liu, Zhihong, et al. "2D-Shapley: A Framework for Fragmented Data Valuation." ICML. 2023.
>
> ----
>
> ### **Question-2: Difference between Trojan and BadNets trigger**
>
> **Re.:** BadNets attack is implemented by directly inserting a white square into the original images. In contrast, Trojan attack specifically targets a pre-trained neural network model. The Trojan trigger is generated by initializing the trigger as a white square and tuning the pixel values in the square to strongly activate some selected neurons in the neural network [1]. We believe the similar performance of the two methods on BadNets square is likely due to the trigger acting as a significant feature. In other words, the trigger pixels significantly differ from the other pixels, making it easily detectable by both methods. The result on Trojan square suggests that 2D-OOB has a stronger detection capability against non-random outliers in applications compared to 2D-KNN, as KNN algorithms focus on local proximity and can fail to effectively capture global patterns in feature behavior. We will include this discussion in the revision.
>
> [1] Yingqi Liu, Shiqing Ma, Yousra Aafer, Wen-Chuan Lee, Juan Zhai, Weihang Wang, and X. Zhang. 352 Trojaning attack on neural networks. In Network and Distributed System Security Symposium, 2018. URL 353 https://api.semanticscholar.org/CorpusID:31806516.
>
> ----
>
> ### **Question-3: Original prediction performance for each task**
>
> **Re.:** The average accuracy (across B=1000 weak learners) of various weak learners are summarized in the table below. There are certain variations across the datasets. It is important to note that the original prediction performance is not correlated with the joint valuation performance.
>
> |Dataset|DecisionTree|LogisticRegression|MLP(single-layer)|MLP(two-layer)|
> |-------|------------|------------------|-----------------|--------------|
> |lawschool|0.78|0.84|0.83|0.83|
> |electricity|0.65|0.67|0.68|0.68|
> |fried|0.64|0.71|0.71|0.71|
> |2dplanes|0.67|0.70|0.71|0.70|
> |creditcard|0.71|0.79|0.80|0.79|
> |pol|0.78|0.74|0.83|0.84|
> |MiniBooNE|0.80|0.74|0.81|0.84|
> |jannis|0.61|0.68|0.66|0.64|
> |nomao|0.88|0.91|0.92|0.91|
> |vehicle_sensIT|0.73|0.78|0.81|0.79|
> |gas_drift|0.98|0.99|1.00|1.00|
> |musk|0.88|0.90|0.93|0.93|
>
> ----
>
> ### **Question-4: The implication of the connection to data-OOB in Proposition 3.1.**
>
> **Re.:** Proposition 3.1 shows that the 2D-OOB $\psi_{ij} ^{\mathrm{2D-OOB}}$ can be interpreted as the empirical expectation of the Data-OOB, conditioned on the inclusion of the j-th feature.
>
> To clarify further, let's consider a simple case: assume that for a given data point, each specific feature subset is selected an equal number of times when it is not selected in a bootstrap dataset, i.e., it is in an out-of-bag. Under this assumption, the point mass of $j$-th cell within the data point is $\frac{1}{\sum_{l=1}^{L}\mathbb{1}(j\in S_l)}$, which uniformly considers all feature coalitions including $j$-th feature.
>
> It justifies why the definition in Eqn. (4) (line 120) effectively attributes a data point's value to features. It shows that for a fixed $i$ and $ j \neq k$, $\psi_{ij} ^{\mathrm{2D-OOB}}>\psi_{ik} ^{\mathrm{2D-OOB}}$​ implies that the cell $x_{ij}$ is more helpful in achieving a high OOB score, serving as an indicator of model performance.

---

> > ### Comment · Reviewer_4K7L · 2024-08-11
> > **Thanks for the response**
> >
> > I appreciate the authors' response. Overall I think this is a solid paper, although its impact may be limited by its applicability to bagging methods only.

---

> > > ### Author Response · Authors · 2024-08-14
> > > **Thank you**
> > >
> > > We appreciate your thoughtful review again and are pleased you found the paper solid!

---

### Official Review · Reviewer_6nYm · 2024-07-15

**Soundness:** 3
**Presentation:** 3
**Contribution:** 2
**Rating:** 5
**Confidence:** 2

**Summary:**

This paper studies data valuation in a fine-grained fashion: determine helpful samples as well as the particular cells that drive them. The proposed method is tested with 12 datasets and has application in backdoor detection.

**Strengths:**

- It makes sense to use cells rather than single data samples as the smallest unit for data valuation.
- The proposed framework is clearly stated.
- The backdoor trigger detection is an interesting and useful application of the proposed framework.
- The proposed method is significantly more efficient than 2D-KNN.

**Weaknesses:**

- The proposed method seems to be a combination of existing algorithms 2D-KNN and Data-OOB. The novelty and contribution may be relatively marginal.

**Questions:**

Could you please discuss what are the *unique* challenges of 2D-OOB that do not arise for 2D-KNN and Data-OOB?

**Limitations:**

Yes.

---

> ### Author Rebuttal · Authors · 2024-08-06
>
> ### **Weakness: The novelty and contribution may be relatively marginal.**
>
> **Re.:** Thank you for your feedback. We acknowledge that 2D-OOB may initially appear to be a natural extension of existing methods, as it is built on them. However, we believe our contribution is substantial in the field of data valuation, as our proposed method addresses both practical and conceptual challenges of 2D-Shapley and Data-OOB.
>
> Compared to 2D-Shapely, our work significantly improves **computational efficiency**, as demonstrated by the empirical experiments in the submitted paper and the new results in our general response. In addition, we identify **new** use cases of the joint valuation framework, such as attributing data contribution through cell values, correcting erroneous cells, and detecting backdoor triggers, which have not been discussed in 2D-Shapley.
>
> In comparison with Data-OOB, which only provides data-level values, our method is designed to assess cell-level values. This finer granularity in valuation presents a richer interpretation of data values by providing **attribution for data values**. Also, it enables new practical applications.
>
> ----
>
> ### **Question: Could you please discuss what are the unique challenges of 2D-OOB that do not arise for 2D-KNN and Data-OOB?**
>
> **Re.:** The main technical contribution of 2D-OOB lies in the adoption of a **subset bagging model**, rather than the standard bagging model used in Data-OOB. This requires more careful formulation (as detailed in Equation 4) and presents additional challenges, such as more complex hyperparameter selection (for example, the feature subset ratio, which we have thoroughly examined in the ablation study in Appendix B.4).

---

### Author Rebuttal · Authors · 2024-08-06

We want to thank all reviewers and AC for their dedicated work. We are appreciative that reviewers agreed our work to be **well-motivated, clearly-written, and include extensive experiments**. The concept of joint valuation was recognized **reasonable** (6nYm) and **important** (4K7L). Our proposed framework was considered **easy to follow** (4K7L, nD8c) and **theoretically justified** (ECZV). Reviewers 6nYm, 4K7L and ECZV commended **the algorithm's superior computational efficiency compared to existing methods**. Reviewers 6nYm, 4K7L and nD8c acknowledged the backdoor trigger detection experiment as **useful, interesting and promising**.

As major additions, we have added **a new set of experiments on the scalability of 2D-OOB**.

## Experiment on the scalability of 2D-OOB (Reviewers 4K7L, nD8c)

We evaluate the computational efficiency of our proposed approach using a synthetic binary classification dataset. For $d \in \lbrace 20, 100, 500, 1000 \rbrace$, input data $X \in \mathbb{R}^d$ is sampled from a multivariate Gaussian distribution with a mean of zero and an identity covariance matrix. The output labels Y in $Y \in \{0, 1\}$ are generated using a Bernoulli distribution with a success probability determined by $p(X) := 1/(1 + exp(-X^T\eta))$, where each component of the vector $\eta \in \mathbb{R}^d$ is sampled from a standard Gaussian distribution. We experiment with sample sizes $n \in \lbrace10^3, 10^4, 10^5\rbrace$. We record the elapsed time using the same computational device. The elapsed time for 2D-OOB includes the training time for the subset bagging model to ensure a fair comparison.

As shown in the table below, our 2D-OOB enjoys remarkable scalability, in terms of both sample size n and feature dimension d. We also examined 2D-KNN under the same settings. However, for larger datasets, 2D-KNN required more than 5 hours to complete (which are omitted from the table). Notably, 2D-KNN with n=10000 and d=20 took 7556.32 seconds, which is even slower than our method with n=100000 (10x larger) and d=1000 (50x larger), which took 7423.87 seconds.


| n      | d    | 2D-OOB (ours) | 2D-KNN |
|--------|------|---------------|--------|
| 1000   | 20   | 10.16         |   714.89     |
| 1000   | 100  | 17.35         |   3564.51     |
| 1000   | 500  | 56.67         |   /     |
| 1000   | 1000 | 113.06        |    /    |
| 10000  | 20   | 53.89         |   7556.32     |
| 10000  | 100  | 165.55        |   /     |
| 10000  | 500  | 681.60        |   /     |
| 10000  | 1000 | 1303.33       |  /      |
| 100000 | 20   | 604.38        |   /     |
| 100000 | 100   | 845.83        |      /  |
| 100000 | 500   | 3927.31       |    /    |
| 100000 | 1000   | 7423.87       |    /    |

Table: Runtime (in seconds) for 2D-OOB under different data sizes (n) and input dimensions (d). We omitted the runtime for cases where the computation exceeded 5 hours.

---

### Comment · Area_Chair_idkm · 2024-08-08

Dear reviewers,

All the reviews and rebuttals are now available. Please go over all such information and ask any clarifying questions to the authors.

---

### Decision · Program_Chairs · 2024-09-25

**Decision:**

Accept (poster)

**Comment:**

The rebuttal provided by the authors have addressed most of the reviewers' questions, by clarifying parts of the submission and providing additional experiments. However, some of the reviewers also point out that the novelty of the submission and its applicability may be limited.